# Effect of the 2022 summer drought across forest types in Europe

Mana Gharun[1], Ankit Shekhar[2,3], Jingfeng Xiao[4], Xing Li[5], Nina Buchmann[2]

[1] Institute of Landscape Ecology, University of Münster, Münster, Germany
[2] Institute of Agricultural Sciences, ETH Zürich, Zürich, Switzerland
[3] Agricultural and Food Engineering Department, Indian Institute of Technology Kharagpur, Kharagpur, India
[4] Earth Systems Research Center, University of New Hampshire, New Hampshire, USA
[5] Research Institute of Agriculture and Life Sciences, Seoul National University, Seoul, South Korea

Correspondence to: Mana Gharun, mana.gharun@uni-muenster.de

## Abstract

Forests in Europe experienced record-breaking dry conditions during the 2022 summer. The direction in which various forest types respond to climate extremes during their growing season is contingent upon an array of internal and external factors. These factors include the extent and severity of the extreme conditions and the tree ecophysiological characteristics adapted to environmental cues, which exhibit significant regional variations. In this study we aimed to: 1) quantify the extent and severity of the extreme soil and atmospheric dryness in 2022 in comparison to two most extreme years in the past (2003 and 2018), 2) quantify response of different forest types to atmospheric and soil dryness in terms of canopy browning and photosynthesis, and 3) relate the functional characteristics of the forests to the emerging responses observed remotely at the canopy level. For this purpose, we used spatial meteorological datasets between 2000 to 2022 to identify conditions with extreme soil and atmospheric dryness. We used the near-infrared reflectance of vegetation (NIRv) derived from the MOderate Resolution Imaging Spectroradiometer (MODIS), and the Global OCO-2 Solar Induced Fluorescence (GOSIF) as an observational proxy for ecosystem gross productivity, to quantify the response of forests at the canopy level.

In summer 2022, southern regions of Europe experienced exceptionally pronounced
atmospheric and soil dryness. These extreme conditions resulted in a 30% more
widespread decline in GOSIF across forests compared to the drought of 2018, and 60%
more widespread decline compared to the drought of 2003. Although the atmospheric
and soil drought were more extensive and severe (indicated by a larger observed
maximum z-score) in 2018 compared to 2022, the negative impact on forests, as
indicated by declined GOSIF, was significantly larger in 2022. Different forest types were
affected in varying degrees by the extreme conditions in 2022. Deciduous broad-leaved
forests were the most negatively impacted due to the extent and severity of the drought
within their distribution range. In contrast, areas dominated by Evergreen Needle-Leaf
Forests (ENF) in northern Europe experienced a positive soil moisture (SM) anomaly and
minimal negative vapor pressure deficit (VPD) in 2022. These conditions led to enhanced
canopy greening and stronger solar-induced fluorescence (SIF) signals, benefiting from
the warming. The higher degree of canopy damage in 2022, despite less extreme
conditions, highlights the evident vulnerability of European forests to future droughts.
Keywords: *photosynthesis, soil drought, atmospheric drought, canopy browning, gross*
*primary production*

## Introduction

The frequency and intensity of drought events have been rising globally, and future global
warming is expected to further increase their occurrence (Seneviratne et al. 2021;
Röthlisberger and Papritz 2023). Particularly over the past two decades, many regions in
Europe have experienced widespread drought conditions, notably during the summers of
2003, 2010, and 2018 (Bastos et al. 2020; Zhou et al. 2023). The extreme conditions
caused widespread ecological disturbances (Müller and Bahn 2022) and reduced the
capacity of forests for carbon uptake, thereby diminishing their potential for mitigating
climate change (van der Woude et al. 2023). Additionally, heatwaves and prolonged
droughts stress vegetation, making it more susceptible to other biotic and abiotic stress
factors. This increased vulnerability leads to higher tree mortality, elevated wildfire risks,
and a loss of biodiversity among plants and animals living at the edge of their temperature
tolerance. These conditions also alter phenology and plant development, causing
cascading effects on ecosystem functioning (Seidl et al. 2017).
The spatial extent and severity of drought events vary, and their impacts depend on local
ecological characteristics of the forests, species-specific temperature and moisture
thresholds that limit tree functioning, as well as adaptation strategies and acclimation of
trees to more frequent and intense extreme conditions (Gessler et al. 2020). For example,
comparing the 2003 and 2018 extreme years, the year 2018 was characterized by a
climatic dipole, featuring extremely hot and dry weather conditions north of the Alps but
comparably cool and moist conditions across large parts of the Mediterranean. Negative
drought impacts appeared to affect an area 1.5 times larger and to be significantly
stronger in summer 2018 compared to summer 2003 (Buras et al. 2020).
In 2022, Europe faced its second hottest and driest year on record, with the summer of
that year being the warmest summer ever recorded. Conditions in summer 2022 led to
record-breaking heatwave and drought events across many regions (Copernicus Climate
Change Service, 2023). Compound drought and heatwave conditions in 2022 caused
widespread crop damage, water shortages, and wildfires across Europe. The hardest-hit
areas were the Iberian Peninsula, France, and Italy, where temperatures exceeded 2.5°C
above normal, and severe droughts persisted from May to August (Tripathy and Mishra
2023). The reduced soil moisture due to precipitation deficits and high temperatures,
contributed to the persistence and severity of drought, creating a positive feedback loop
where dry soils led to even drier conditions (Tripathy and Mishra 2023).
Drought and heatwaves have a range of detrimental effects on trees and forests. The
most immediate impact is that elevated air temperatures and increased dryness, whether
in the soil or in the atmosphere, disrupt mesophyll and stomatal conductance, thereby
impairing carbon uptake (Marchin et al. 2022). Plants reduce stomatal conductance under
severe drought to reduce water stress at the expense of reduced rates of photosynthesis
(Oren et al., 1999). Drought also increases the chance of hydraulic failure, which can lead
to tree mortality (Choat et al. 2018). Additionally, rising temperatures reduce the
enzymatic activity in trees, which in turn diminishes the forest´s gross primary productivity
(Gourlez de la Motte et al. 2020). Elevated temperatures can also increase respiration
rates in both soil and trees, which reduces the forest´s net carbon uptake and their ability
to mitigate anthropogenic $CO_2$ emissions (van der Molen et al. 2011; Anjileli et al. 2021).
Drought also restricts the movement of nutrients in soil water, reducing their availability
to trees and consequently impacting their growth and productivity (Bauke et al. 2022).
Changes in plant water-use and nutrient cycling can trigger feedback loops that magnify
the effects of drought and heat stress. For instance, reduced plant cover can increase
soil temperatures and further accelerate water loss and increase plant water demand
(Haesen et al. 2023). On the other hand, increased atmospheric dryness or reduced soil
moisture levels increase stomatal closure which limits transpiration and leads to higher
leaf temperature that intensifies heat stress on plants (Drake et al. 2018). Reduced
transpiration and photosynthesis elevate surface temperatures and atmospheric $CO_2$
concentrations, altering local and regional climate patterns and intensifying the frequency
and severity of extreme events (Humphrey et al. 2018). These effects vary significantly
depending on forest type and species composition. Together with the characteristics of
the extreme events themselves – such as their extent and severity- this variability
complicates our understanding of how drought affects the functionality of different forest
ecosystems (Gharun et al. 2020; Shekhar et al. 2023). These feedback loops highlight
the urgent need to assess how climate extremes impact different forest types, which are
crucial for sequestering significant portions of anthropogenic emissions. Our study aims
to 1) quantify the extent and severity of the extreme conditions in 2022 – focusing on soil
and atmospheric dryness- and compare them to those of two previous extreme years
(2003, 2018), 2) quantify the responses of different forest types to drought in terms of
canopy browning and photosynthesis, and 3) connect the functional characteristics of the
forests with the canopy-level responses observed.

## Methods


*Meteorological dataset*
We used Europe-wide gridded datasets covering daily mean air temperature (Tair; °C),
daily mean relative humidity (RH; %) and daily mean soil moisture (SM; $m^3m^{-3}$) for the
topsoil layer (0-7 cm depth), spanning from 2000-2022. The study area encompasses
longitudes from 11°W to 32°E, and latitudes from 35.8°N to 72°N, approximately 4.45
million km$^2$. We sourced the Tair and RH datasets from the E-OBS v27.0e dataset which
provides daily data at 0.1°✕0.1° spatial resolution (Cornes et al., 2018; Klein et al., 2002).
We calculated daily mean vapor pressure deficit (VPD; kPa) from Tair and RH using
Equation 1 (Dee et al. 2011).

$$VPD = (1 - \frac{RH}{100}) \times 0.6107 \times 10^{\frac{7.5 \times Tair}{237.3 + Tair}} \quad (1)$$

The topsoil SM dataset was extracted from the most recent reanalysis data from
ECMWF's (European Centre for Medium-range Weather Forecasts) new land component
of the fifth generation of European Reanalysis (ERA5-Land) dataset (daily at 0.1°✕0.1°
resolution; Munoz-Sabater et al., 2021). ERA5-Land provides soil moisture (SM) data at
an hourly interval with a spatial resolution of 0.1° × 0.1°. For our analysis, we aggregated
the hourly SM data into daily averages. Recent validation studies using in-situ
measurements and satellite data have confirmed the high accuracy of surface SM
simulations from ERA5-Land (Albergel et al., 2012; Lal et al., 2022; Muñoz-Sabater et al.,
2021). Additionally, SM data from ERA5-Land have been utilized to investigate drought
and global SM patterns (see Lal et al., 2023; Shekhar et al., 2024b).  We re-sampled the
Tair, VPD, and SM data from daily (0.1° × 0.1°) to 8-day (0.05° × 0.05°) intervals to align
with the temporal and spatial resolution of the vegetation response dataset (see below).
*Forest canopy response dataset*
In order to assess the forest canopy response to drought stress, we used two satellite-
based proxies:
1) The structure-based NIRv (Near-Infrared Reflectance of Vegetation) index derived
from MODIS (Moderate Resolution Imaging Spectroradiometer; 8-day 500m x 500m
MOD09Q1 v6.1 product) which is calculated using surface spectral reflectance at near-
infrared band ($R_{NIR}$) and red band ($R_{Red}$) as shown in Equation 2 (Badgley et al. 2017).
The calculated NIRv at 500m resolution was aggregated to a 0.05°✕0.05° resolution
(daily) by averaging.

$$NIR_V = R_{NIR} \times \frac{R_{NIR} - R_{Red}}{R_{NIR} + R_{Red}} \qquad (2)$$

2) The physiological-based reconstructed global OCO-2 (Observation Carbon
Observatory - 2) solar induced fluorescence (GOSIF) dataset. Solar-induced
fluorescence (SIF) is an energy flux (unit: $Wm^{-2}\mu m.sr^{-1}$) re-emitted as fluorescence by the
chlorophyll *a* molecules in the plants during photosynthesis (Baker, 2008). Recent
extensive research has established a strong link between Solar-Induced Fluorescence
(SIF) and vegetation photosynthesis, validating SIF as an effective proxy for ecosystem
gross primary productivity (GPP) (Li et al. 2018; Magney et al. 2019; Shekhar et al., 2022).
The GOSIF dataset was created by training a Cubist Regression Tree model to gap-fill
SIF retrievals from OCO-2 satellite. This was done using MODIS Enhanced Vegetation
Index (EVI) and meteorological reanalysis data from MERRA-2 (Modern-Era
Retrospective analysis for Research and Applications), which includes photosynthetically
active radiation (PAR), VPD, and air temperature (see Li and Xiao, 2019). We
downloaded GOSIF data set (v2) from the Global Ecology Data Repository
(http://data.globalecology.unh.edu/data/GOSIF_v2/, last accessed on 25 July 2024). The
GOSIF was available from 2000-2022 at 8-day temporal scale with a spatial resolution of
0.05°✕0.05° (Li and Xiao, 2019).
GOSIF signals provide information about physiological response of forest photosynthesis
while NIRv (a recently developed vegetation index) signals provide information about the
health status of the canopy. NIRv is preferred over NDVI and EVI as it can isolate the
vegetation signal, mitigate mixed-pixel issue, and partly address the influences of
background brightness and soil contamination (Zhang et al. 2022). The two vegetation
proxies used in this study are anticipated to offer complementary insights into vegetation
response to drought.
*Land cover dataset*
In this study, we focused on five different types of forests (and woodlands) across Europe,
namely, evergreen needleleaf forest (ENF), evergreen broadleaf forest (EBF), deciduous
broadleaf forest (DBF), mixed forest (MF), and woody savannas (WSA). The spatial
distribution of the five different forest types across Europe is shown in Figure 1. We used
the yearly MODIS land cover product (MCD12C1 version 6.1 at 0.05°✕0.05° resolution)
for the years of 2001, 2006, 2011, 2016 and 2021, to extract total areas covered by each
forest type. Area of each grid cell was calculated using trigonometric equations
considering the latitudinal and longitudinal variations arising due to Earth's spherical
shape (Ellipsoid). Only areas that were consistently identified as each forest type over the
five-year period were included in the analysis. This means that only pixels common
across these five years were selected, and with more than 50% of the 0.05°✕0.05° pixel
area identified as forests. The forested areas selected for this study encompassed
907,875 km², which represents approximately 24% of Europe's total land area. Out of the
total area about 23% (206´212 km$^2$) was dominated by ENFs distributed largely across
Northern Europe (NEU). Approximately 1% (7´000 km$^2$) of the area was dominated by
EBFs, located entirely in Mediterranean Europe (MED), and about 10% (92´209 km$^2$) was
dominated by DBF which was largely distributed across MED. Approximately 20%
(174´934 km$^2$) of the total forested area was dominated by MFs largely dominating Central
Europe (CEU), and about 47% (427´529 km$^2$) was dominated by WSA mostly found in
NEU (Figure 1).
*Drought detection and statistical data analysis*
The focus of our analysis was on the summer months during three extreme years of 2003,
2018, and 2022. For this purpose, we subset VPD, soil moisture (SM), and both
vegetation proxies (NIRv and GOSIF) for the months of June, July, August (JJA) which
consisted of fourteen 8-day periods, for each forested pixel between 2000 and 2022.  We
restricted our analysis to the months of June-July-August so our study is 1) comparable
with existing studies focused on the summer drought 2) to capture the peak of the warm
and dry conditions across Europe, that would be most stressful for the vegetation
functioning, from the perspective of heat and water supply.
To account for the impact of the observed greening trend across Europe on vegetation
proxy anomalies during the extreme years (2003, 2018, 2022), we applied a detrending
process to the summer mean NIRv and GOSIF data. This detrending was performed
pixel-wise from 2000 to 2022 using a simple linear regression model (Buras et al., 2020).
We then calculated pixel-wise standardized summer anomalies, expressed as z-scores
($Var_z$), for all variables—VPD, SM, and the detrended NIRv and GOSIF (hereafter
referred to as NIRv and GOSIF)—for each year, including the extreme years, using
Equation 3.

$$Var_z\ (unitless)\ = \frac{Var - Var_{mean}}{Var_{sd}} \qquad (3)$$

where, $Var_{mean}$ and $Var_{sd}$ are mean and standard deviation of any variable over the 2000-
2022 period.

In drought identification studies, classification of 'normal' (not to be confused with normal
distribution), 'drought' (used synonymously with 'dry'), or 'wet', is largely done using a
standardized index, such as SPI (Standardized Precipitation Index), SPEI (Standardized
Precipitation Evapotranspiration Index), and z-score among others (see Mishra and
Singh, 2011). All studies that use a standardized index for classification, classify "normal"
conditions when the index is between -1 and 1, and "below normal" conditions when the
index is < -1, and "above normal" conditions when the index > 1 (Jain et al., 2015, Wable
et al., 2019, Dogan et al., 2012, Tsakiris and Vangelis, 2005). In this study, we classified
drought conditions as occurring when soil moisture is below normal (SMz < -1) and VPD
is above normal (VPDz > 1), indicating both soil AND atmospheric dryness. This
threshold-based approach using standardized anomalies aligns with established methods
for drought identification and is pertinent for studying drought impacts on forests. Both
soil moisture and VPD directly affect vegetation functioning, making them effective
proxies for identifying environmental constraints on plant physiological performance.
Furthermore, such classification of 'normal' (and thus, 'above normal' and 'below normal'
used in this study) based on z-scores (also called standardized anomalies) can be done
for any meteorological and/or response variables, such as NIRv and GOSIF done in this
study, making the narration of results coherent across different variables.
We used the Pearson correlation coefficient (*r*) and partial correlation coefficients (Pr) to
understand the spatial (across space for each year) and temporal (during each year)
correlation of GOSIF and NIR$_v$ anomalies with SM and VPD anomalies (Dang et al.,
2022). We calculated the partial correlation coefficient using equations 4-7:

$$Pr(GOSIF, SM) = \frac{r(GOSIF,SM) - r(GOSIF,VPD) \times r(SM,VPD)}{\sqrt{1 - r(GOSIF,VPD)^2} - \sqrt{1 - r(SM,VPD)^2}} \qquad (4)$$

$$Pr(GOSIF, VPD) = \frac{r(GOSIF,VPD) - r(GOSIF,SM) \times r(SM,VPD)}{\sqrt{1 - r(GOSIF,SM)^2} - \sqrt{1 - r(SM,VPD)^2}} \qquad (5)$$

$$Pr(NIRv, SM) = \frac{r(NIRv,SM) - r(NIRv,VPD) \times r(SM,VPD)}{\sqrt{1 - r(NIRv,VPD)^2} - \sqrt{1 - r(SM,VPD)^2}} \qquad (6)$$

$$Pr(NIRv, VPD) = \frac{r(NIRv,VPD) - r(NIRv,SM) \times r(SM,VPD)}{\sqrt{1 - r(NIRv,SM)^2} - \sqrt{1 - r(SM,VPD)^2}} \qquad (7)$$
**Results**
*Severity of the 2022 summer drought compared to 2018 and 2003*
Figure 2 shows the extent and magnitude of anomalies (z-score) of VPD and top layer (0-
7 cm) soil moisture content during the summer months in 2003, 2018, and 2022 across
Europe. In summer 2022, particularly southern regions of Europe experienced the most
pronounced increase in atmospheric (z-score > 1) and soil dryness (z-score < -1) (Figure
2) while in 2018 we observed the most widespread VPD and SM anomalies in northern
Europe (Figure 2).
Figure 3 shows the intensity of atmospheric and soil drought via z-score values of VPD
and SM anomalies over the summer months (JJA) in 2003, 2018, and 2022. The total
affected area displayed in Figure 3 is the sum of all pixels within the given z-score bin
during the summer period where z-scores are averaged for each bin for the summer
period. Restricted to forested areas, atmospheric and soil drought was 55% and 58%
more extensive in 2018 compared to 2022, and in both years more extensive than in 2003
(Figure 3). In 2022, 28 Mha of forested areas in Europe experienced an extremely high
VPD (z-score > 1), while in 2018, 63 Mha experienced such extreme conditions. In 2022,
21 Mha of forested areas experienced an extremely low soil moisture content (z-score <
-1) while in 2018, 50 Mha of forests in Europe were affected by such extreme conditions.
In 2003 an area of 25 Mha was affected by extremely dry air and a similar area was
affected by extremely dry soil (Figure 3). A comparison of soil drought detected from SM
at 0-100 cm showed a similar result in terms of drought severity and spatial coverage and
thus we used SM at 0-7 cm soil layer for our analysis (see Supplementary Figure 1).

*Forest canopy response to the 2022 drought*
The intensity of GOSIF and NIRv anomalies over the summer months (JJA) in 2003,
2018, and 2022 are displayed in Figure 4. The extent shown in Figure 4 is the sum of all
pixels within the given z-score bin during the summer period (z-scores are averaged for
each bin). Compared to 2018, the extremely dry conditions in 2022 led to 30% increase
in forested areas that exhibited declined photosynthesis (17 Mha in 2022 compared to 12
Mha in 2018) (Figure 4). The extent of the canopy browning observed in 2022 was similar
to 2018, which in both years was 120% of the extent of observed canopy browning in
2003 (11 Mha compared to 5 Mha observed in 2003) (Figure 4).
Figure 5a shows the GOSIF anomalies (z-score) across all forested areas in Europe. The
intensity and extent of the GOSIF anomalies during the summer months (JJA) in each
year are shown for different forest types in Figure 5b. Across specific forest types, DBFs
showed the largest negative GOSIF anomaly in 2022 but the ENFs showed a positive
GOSIF anomaly in 2022, both in terms of magnitude and in terms of the spatial extent of
negative GOSIF anomalies (Figure 5).
Figure 6a shows the anomalies of NIRv (average z-score over the summer months)
across all forested areas in Europe. The intensity and extent of the NIRv anomalies during
the summer months (JJA) in each year are shown for different forest types in Figure 6b.
In terms of canopy browning response (NIRv anomalies), the largest negative NIRv
anomalies in 2022 were observed in southern Europe (Figure 6). Largest negative NIRv
anomalies (indicated by the maximum anomaly) were observed in the DBFs in 2022,
fitting the declined GOSIF signals. The ENFs showed positive NIRv anomalies in 2022,
in terms of magnitude, spatial coverage, and % of total area affected (Figure 6).

*Relationship between GOSIF and NIRv*
In general, the values of NIRv and GOSIF were highly correlated (Supplementary Figure
2). The anomalies of NIRv and GOSIF were most correlated across WSAs ($r^2$ = 0.73 in
2018) and least correlated across the ENFs (Supplementary Figure 2). Figure 7 shows
the spatial regression between standardized GOSIF anomalies with (a) VPD and (b) SM
and Figure 8 shows the spatial regression between standardized NIRv anomalies with (a)
VPD and (b) SM over the drought areas in summers 2003, 2018 and 2022. With the
increase in VPD (i.e., increased atmospheric dryness), GOSIF values declined across all
forest types, across all years, except in 2022 in the WSA, and in 2018 and 2022 in EBFs
(Figure 7). With decrease in soil moisture (i.e., increased soil dryness), GOSIF values
also declined overall ($r^2$ = 0.34), but not as strongly as with the increase in air dryness ($r^2$
= 0.39) (Figure 7). Across different forest types, GOSIF responded most strongly to VPD
anomalies in the MFs (mean $r^2$ = 0.48), and responded most directly to changes in the
soil moisture in the WSA (Figure 7).
Between VPD and SM, in general GOSIF anomalies were more correlated with VPD than
with SM anomalies, and the decline in VPD correlated well with the larger GOSIF decline
that we observed in DBFs in 2022 and in ENFs in 2003 (Figure 7). Under typical
conditions (regardless of drought), GOSIF's response to both air dryness and soil
moisture anomalies was more pronounced than the response of NIRv ($r^2$ = 0.39 with
GOSIF, compared to $r^2$ = 0.29 for NIRv) (Figure 7, 8).
Figure 9 shows the partial correlation coefficient between GOSIF with SM and VPD during
summer months (JJA) for areas identified as affected (Figure 9a) and not affected (Figure
9b) by drought. The SM and VPD values across all forest types correlated well, but across
DBFs the dryness in the atmosphere and the dryness in the soil were most correlated
(Figure 9). Regarding canopy response to VPD, European Needleleaf Forests (ENF)
exhibited the strongest reaction to changes in atmospheric dryness (Figure 9)

## 321    Discussion

*Severity of the 2022 summer drought*
Although the years 2003, 2018, and 2022 are all categorized as "extreme," the specific
characteristics of the extreme conditions varied significantly among these years. For
example, in 2003, widespread negative anomalies in soil moisture signaled a significant
soil drought, whereas in 2022, widespread positive VPD anomalies indicated a notably
drier atmosphere (Figure 3). It is important to note that ERA-5 Land datasets have been
shown to underestimate the extent of European heatwaves in 2003, 2010, and 2018
(Duveiller et al., 2023), partly due to the use of a static leaf area index in their modeling
framework. Consequently, the SM droughts in the years 2003, 2018, and 2022 may be
more severe than indicated by our study, suggesting that our results might be somewhat
conservative. The extensive summer drought in 2022 primarily impacted southern
Europe, in contrast to the 2003 summer drought, which affected central Europe, and the
2018 drought, which extended to central and northern Europe (Figure 2) (Bastos et al.,
2020). Consequently, the severe dry conditions in 2022 resulted in an average decline in
GOSIF across forests that was 30% more widespread compared to 2018, and 60% more
widespread compared to 2003 (Figure 4). These above-normal dry conditions during the
summer reduced the photosynthetic capacity of plants and, consequently, the
ecosystem's ability to absorb carbon from the atmosphere (Peters et al., 2018; van der
Woude et al., 2023). Although the atmospheric and soil droughts in 2018 were more
extensive and severe compared to 2022 (as indicated by the maximum observed z-
scores), the adverse impact on forests, as reflected by the decline in GOSIF, was greater
in 2022.
*Canopy response to soil versus atmospheric dryness*
The GOSIF dataset used in this study has been shown to be a reliable proxy for
vegetation gross productivity, as demonstrated by comparisons with ground-based flux
measurements (Shekhar et al. 2022; Pickering et al. 2022). It is important to note that
GOSIF estimates are derived from a machine learning model trained with OCO-2 SIF
observations, MODIS EVI data, and meteorological reanalysis data. As a result, the
meteorological data used in our analyses are not entirely independent of the SIF data.
However, this overlap is unlikely to impact our findings. A recent study that compared
GOSIF with original OCO-2 data to assess the impacts of the 2018 U.S. drought found
similar responses to drought between the two datasets (Li et al., 2020).
NIRv and SIF signals are well-correlated and effectively capture seasonal patterns in GPP
(Getachew Mengistu et al. 2021). Although the strength of their relationship can vary with
time, location, and forest type (see Supplementary Figure 2), reductions in SIF signals
are directly associated with decreased photosynthesis. While both SIF and NIRv are
reliable indicators of canopy responses to extreme climate events, SIF is more responsive
to short-term climatic changes (Figure 7).
Our analysis showed that across different regions, GOSIF anomalies corresponded more
strongly to increased atmospheric dryness than to increased soil dryness (Figure 7). This
supports the understanding that vapor pressure deficit plays a larger role in controlling
SIF signals for trees over shorter time scales than soil moisture (Pickering et al. 2022).
Over shorter time frames, trees can often mitigate soil moisture deficits through
mechanisms within the rooting zone and by accessing deeper water sources, whereas
there is no such buffer for the impact of atmospheric dryness on tree canopies.
Ground-based observations in forest ecosystems, including both ecosystem and tree-
level measurements, have shown that atmospheric dryness can constraint canopy gas
exchange, even when soil moisture is not limiting (Gharun et al. 2014, Fu et al. 2022,
Shekhar et al. 2024a). These findings highlight the importance of considering atmospheric
dryness as a limiting factor for tree photosynthesis during extremely dry conditions and
demonstrate the rapid response of various canopy types to increased levels of
environmental dryness.
*Canopy response to drought across different forest types*
The spread of drought, measured as the total area across z-scores, exhibited distinct
patterns in different years, leading to varied responses of different forest types to the
climatic anomalies. Impact of drought on forests can significantly differ depending on the
forest type, tree species, species composition, and past exposure to extreme conditions
(Arthur and Dech 2016; Chen et al. 2022). Our analysis showed that conditions in summer
2022 reduced vegetation functioning across DBFs the most, as it was indicated by
declined GOSIF signals (Figure 5). While deciduous broad-leaved forests were most
negatively affected by the extreme conditions in 2022, Evergreen Needle-Leaf Forests
(ENF) distributed in northern regions of Europe were not exposed to extremely dry
conditions in 2022 and even showed enhanced canopy greening and GOSIF signals,
through benefiting from the episodic warming (Forzieri et al. 2022). Under similar drought
conditions, the mechanisms to cope with the level of drought stress vary largely among
forest types, and depend on a combination of characteristics that control water loss
through the coordination of stomatal regulation, hydraulic architecture, and root
characteristics (e.g., rooting depth, root distribution, root morphology) (Gharun et al. 2020;
Peters et al. 2023). Stomata of trees exhibit a high sensitivity to VPD fluctuations, causing
a reduction in stomatal conductance as VPD increases, which, in turn, limits the exchange
of $CO_2$ with the atmosphere during photosynthesis (Bonal and Guehl in 2011; Li et al.
2023). Different tree species show varying degrees of sensitivity in their stomatal
responses to atmospheric dryness (Oren et al., 1999). For example, ring-porous species
tend to maintain robust gas exchange under dry conditions, while diffuse-porous species,
like those in ENFs, exhibit stronger stomatal regulation, reducing stomatal conductance
as water availability decreases (Klein, 2014). This variability places plants on a spectrum
of drought tolerance, reflecting their specific water relations strategies and leading to
different responses among forests in similar climatic regions.
*Vulnerability of forests to more frequent drought*
The increased canopy damage observed in 2022, despite less severe conditions
compared to the previous extreme year, suggests a lasting impact on forest canopies that
could lead to a decline in forest resilience in the face of more frequent drought events
(Forzieri et al., 2022). A potential decline in the resilience of forests has significant
implications for vital ecosystem services, including the forest's capacity to mitigate climate
change. Consequently, there is an urgent need to consider these trends when formulating
robust forest-based mitigation strategies. This need is especially critical given future
projections indicating that the frequency and intensity of extreme dryness across Europe
will more than triple by the end of the 21st century (Shekhar et al., 2024b). In this context,
it is increasingly important to investigate the vulnerability of forests to external
perturbations and to develop mitigation strategies tailored to site-specific
ecophysiological and environmental factors that influence forest resilience to drought.
Effective management strategies should be based on an understanding of these factors
to mitigate the legacy effects of drought (McDowell et al., 2020; Wang et al., 2023;
Shekhar et al., 2024a).

**Conclusion**
The severity of the 2022 summer drought, marked by increased atmospheric dryness,
significantly compromised the photosynthetic capacity of trees, leading to widespread
declines in vegetation functioning, especially in deciduous broad-leaved forests. Our
findings underscore the importance of considering atmospheric dryness as a critical factor
influencing canopy responses during extreme climatic events, alongside soil moisture
deficits. Despite less severe overall conditions compared to previous extreme years, the
greater canopy damage observed in 2022 suggests a growing vulnerability of forests to
drought. This raises concerns about the future climate mitigation capacity of forest
ecosystems, particularly as projections indicate a continued increase in the frequency and
intensity of extreme dryness across Europe.

**Competing interests**
The authors have no competing interests to declare.

**Financial support**
AS acknowledges funding from the SNF funded project EcoDrive (IZCOZ0_198094). JX
was supported by the National Science Foundation (NSF) (Macrosystem Biology &
NEON-Enabled Science program: DEB-2017870).
**Data availability** The R scripts used for the data analyses and plots are available upon
request from the corresponding author.
**Author contributions** MG, AS, NB conceptualized the study. AS, JX, XL: data
processing. MG and AS: data analyses. MG, AS, JX, XL: paper writing, revision and
editing of the paper.

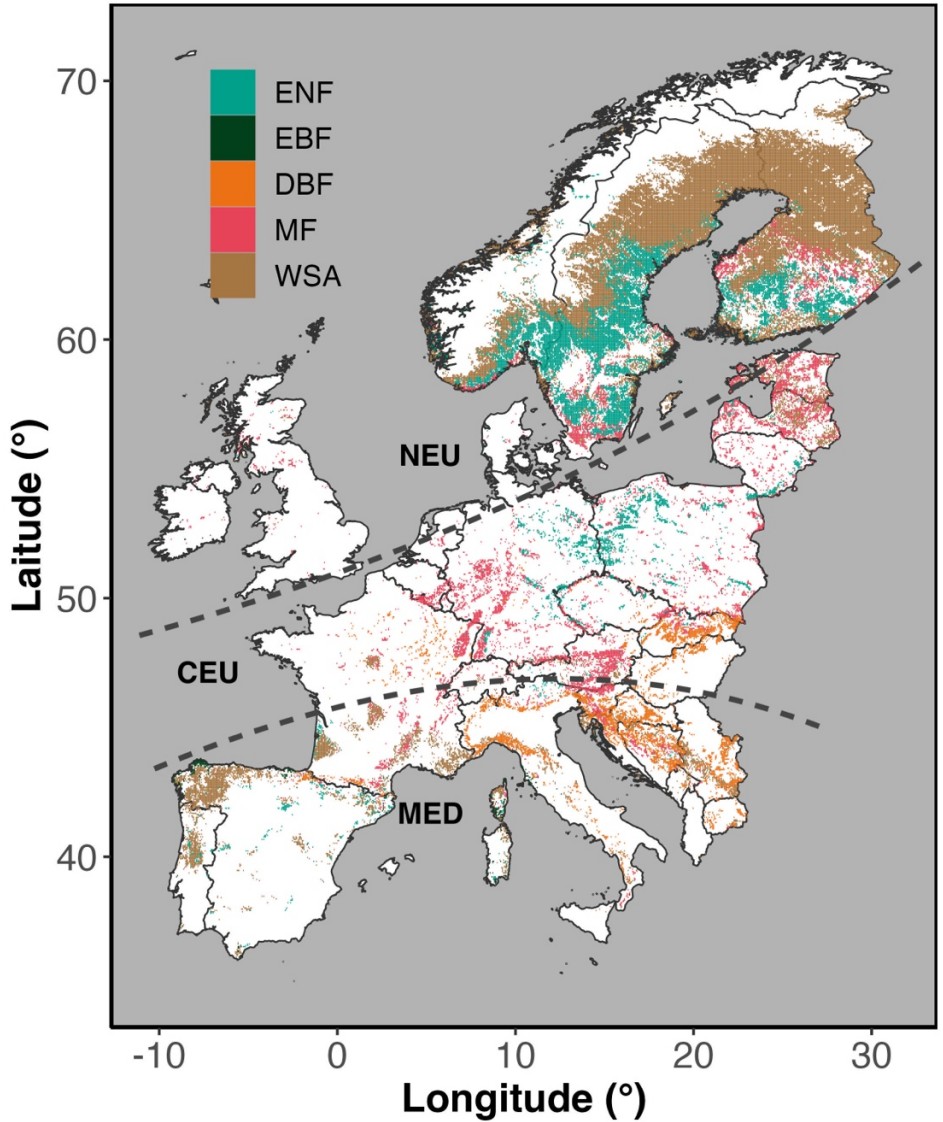

**Figure 1** Spatial coverage of forests (ENF - evergreen needleleaf forest; EBF - evergreen broadleaf forest; DBF - deciduous broadleaf forest; MF - mixed forest), and woodlands (WSA - woody savannas) across Europe. Areas are differentiated into Northern Europe (NEU), Central Europe (CEU), and Mediterranean Europe (MED) following Markonis et al. (2021). The map is based on MODIS land cover product MCD12C1 (version 6.1).

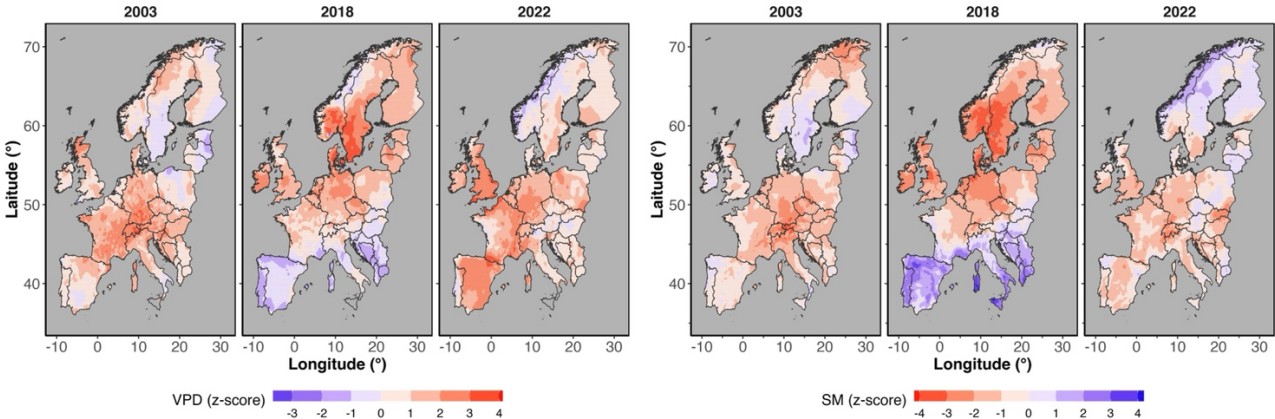


**Figure 2** Standardized summer (JJA) anomalies (z-score) of mean vapor pressure deficit
(VPD), and top layer (0-7 cm depth) soil moisture (SM) in 2003, 2018 and 2022, across
the region of Europe.



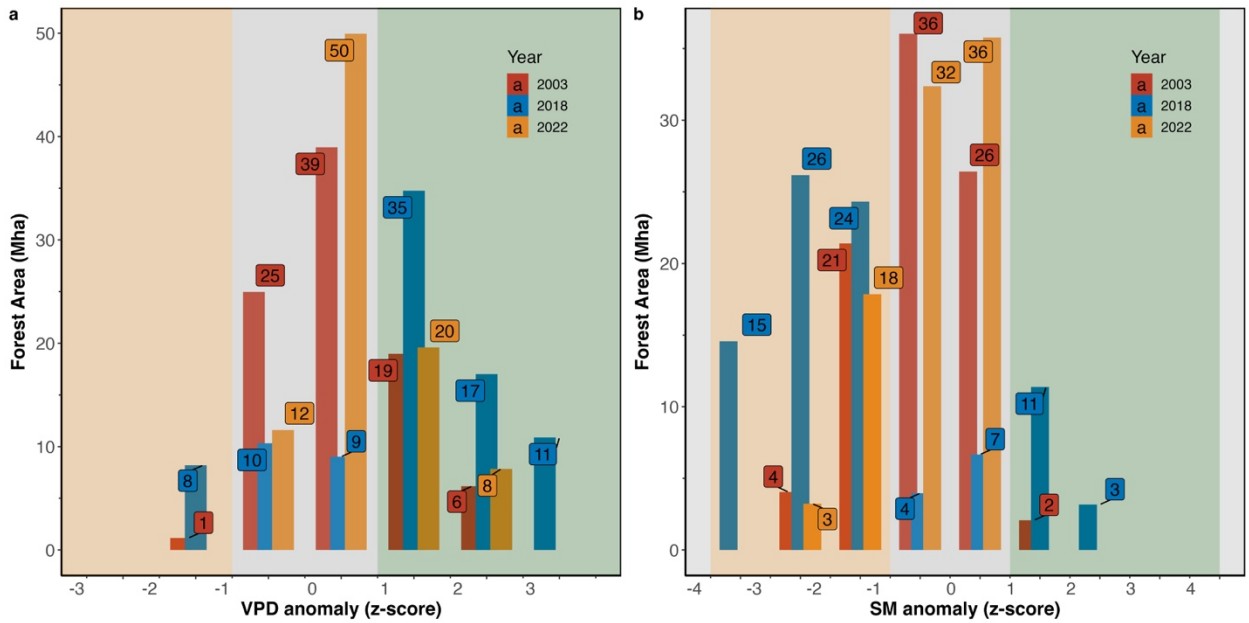



**Figure 3** Intensity (z-score) and extent (area affected, Mha) of (a) VPD, and (b) SM
anomalies across forested areas during the summer months (JJA). Z-score, values from
-1 and 1 are considered normal (within 1 standard deviation of the mean). Orange-shaded
area marks below normal and green-shaded area marks above normal conditions.

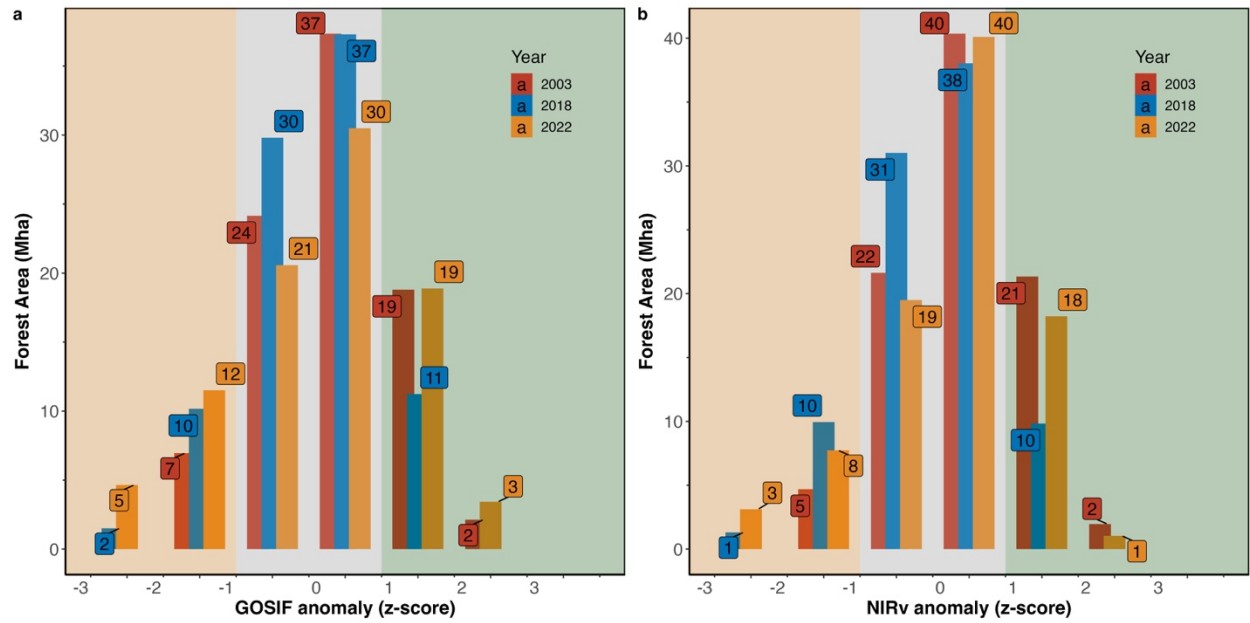

**Figure 4** Intensity (z-score) and extent (area affected, Mha) for (a) GOSIF, and (b) NIRv anomalies across forested areas during the summer months (JJA). Z-score, values from -1 and 1 are considered normal (within 1 standard deviation of the mean). Orange-shaded area marks below normal and green-shaded area marks above normal conditions.

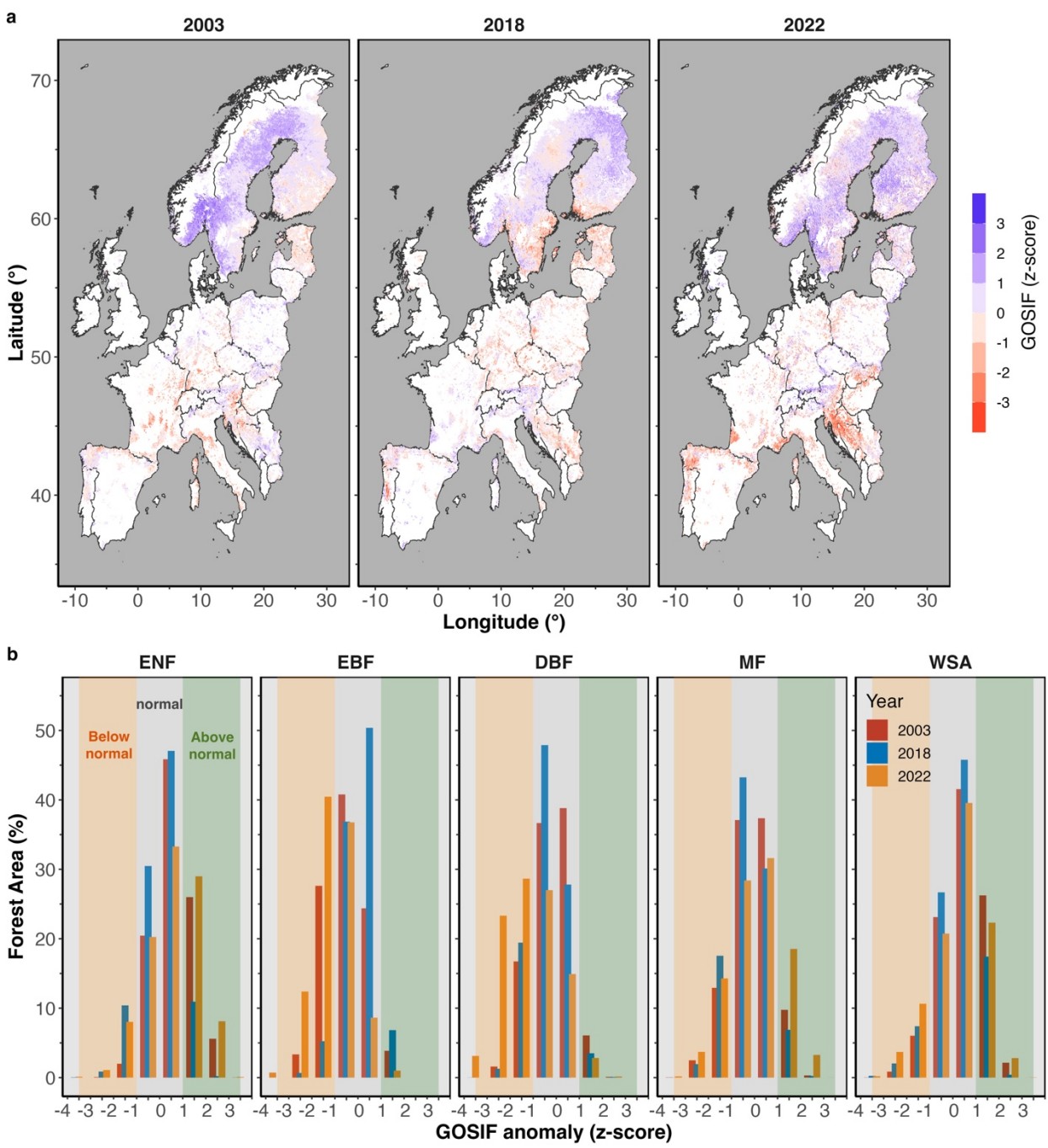

467

**Figure 5** (a) GOSIF anomaly (in terms of z-score) across Europe, and (b) area coverage (in terms of percentage of total area for each forest type) during the summer months (JJA) in 2003, 2018 and 2022. Orange-shaded area marks below normal and green-shaded area marks above normal conditions. White areas on the map mark non-forested regions.

472

473

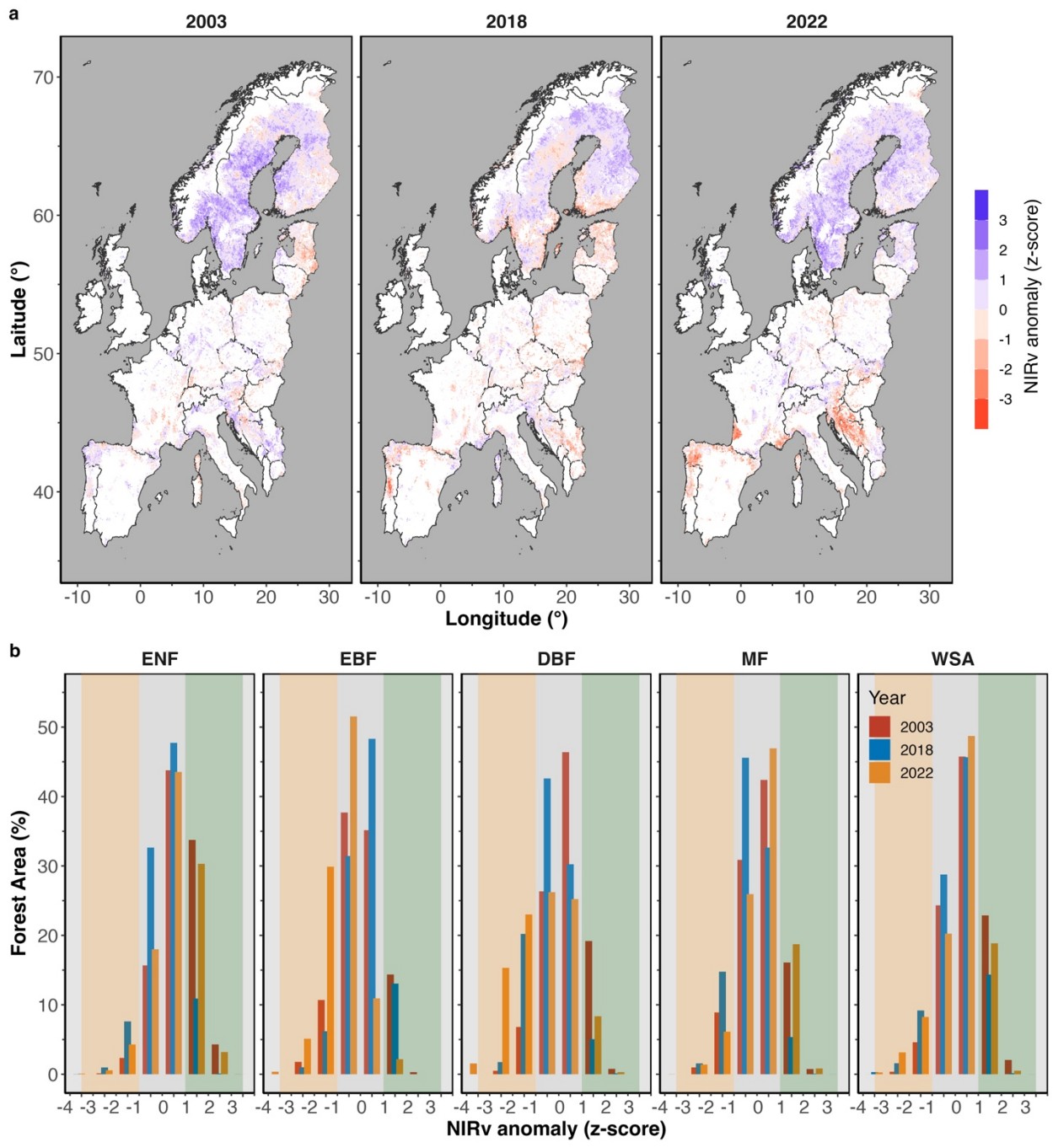

**Figure 6** (a) NIRv anomaly (in terms of z-score) across Europe, and (b) area coverage (in terms of percentage of total area for each forest type) during the summer months (JJA) in 2003, 2018 and 2022. In panel (b) Orange-shaded area marks below normal and green-shaded area marks above normal conditions. White areas on the map mark non-forested regions.

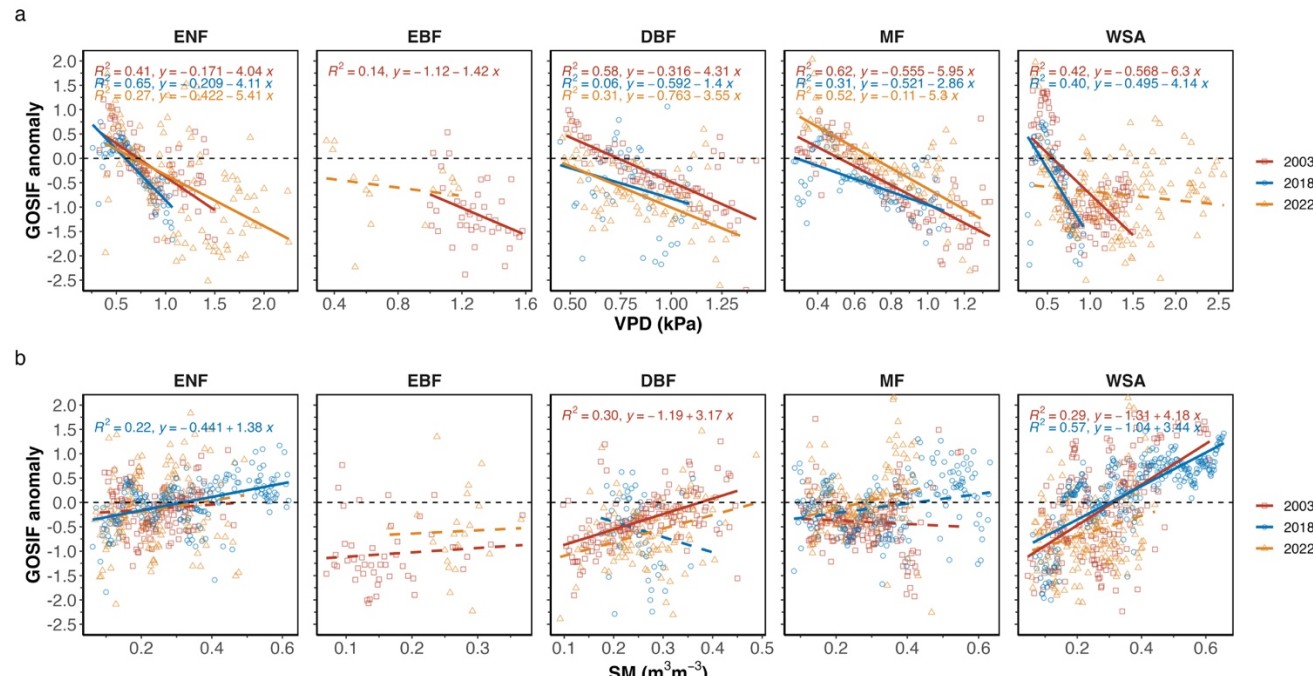


**Figure 7** Spatial regression between standardized GOSIF anomalies with (a) VPD and
(b) SM over the drought areas during the summer months (JJA) 2003, 2018 and 2022.
Dashed lines mark a non-significant relationship ($p > 0.05$).



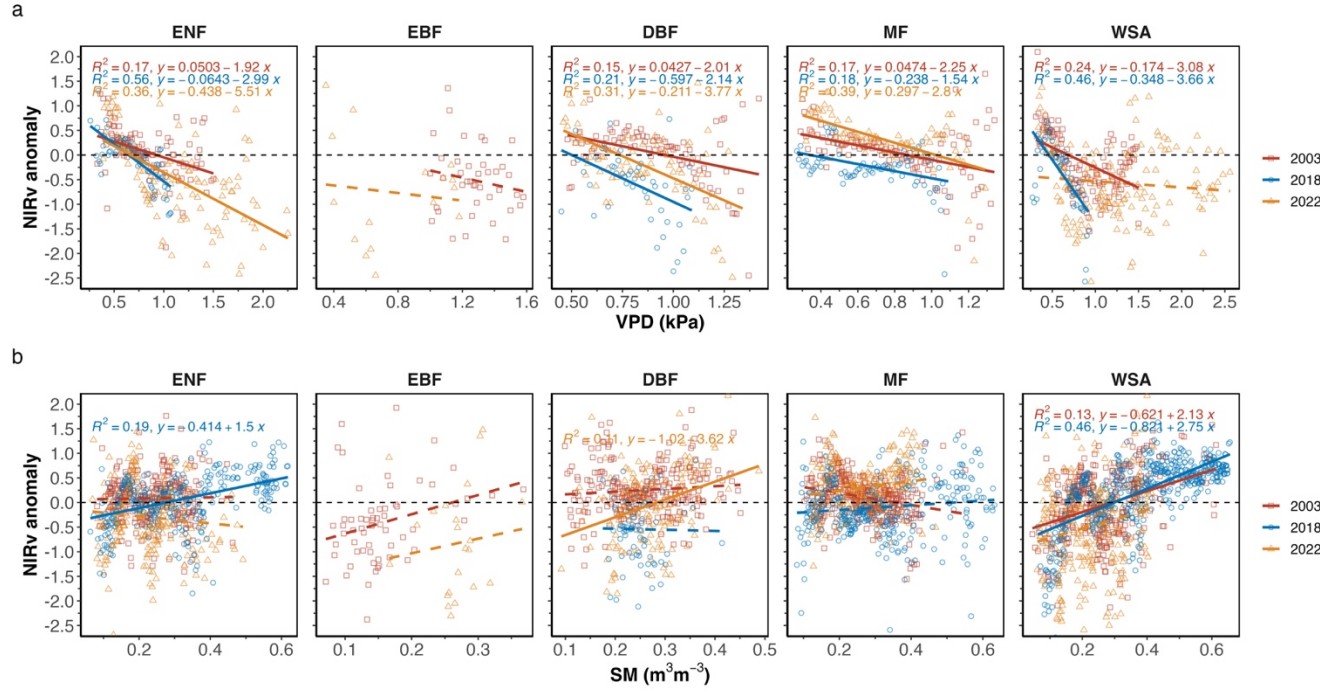



**Figure 8.** Spatial (over all pixels) regression between standardized NIRv anomalies with
(a) VPD and (b) SM over the drought areas and normal areas in 2003, 2018, and 2022
during the summer months (JJA).




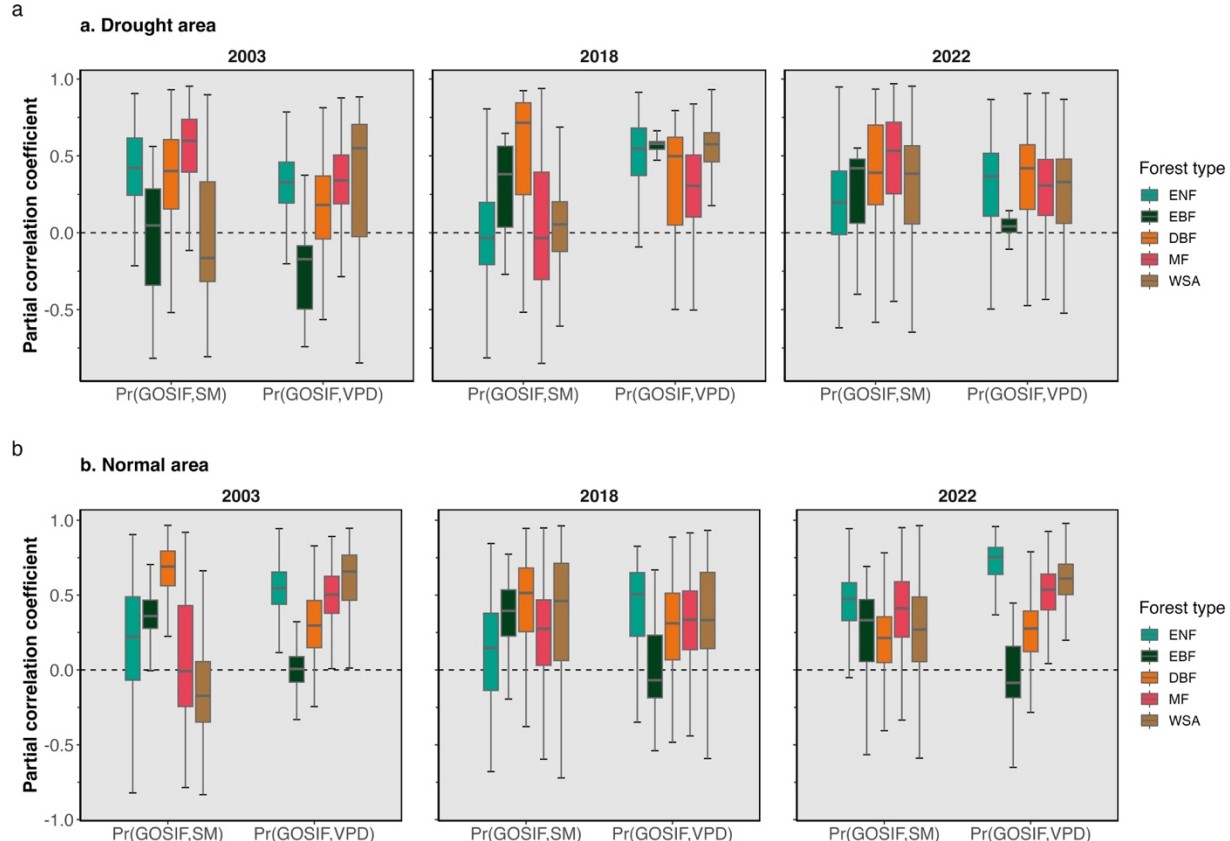


**Figure 9.** Temporal partial correlation coefficient of GOSIF with the absolute values of SM and VPD during the summer months (JJA) in 2003, 2018 and 2022, for detected (a) drought areas and (b) normal areas. A comparable figure for NIRv can be found in Supplementary Figure 3.

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
