# Peer review of "Effect of the 2022 summer drought across forest types in Europe"

_EGUsphere, 2024_

## Author Comment (AC1)

**Discussion of "Effect of the 2022 summer drought across forest types in Europe"**
Gharun et al.

Reviewers' comments are in italic. The Authors´ responses are marked in blue.

**Author Response to Referee 1**

*This study explores the impacts of the 2022 summer droughts on the forest canopy across different European land covers, utilizing two remote-sensing vegetation datasets: near-infrared reflectance of vegetation (NIRv) and solar-induced fluorescence (SIF). The authors analyzed the extent of forests affected and examined the correlations between canopy condition anomalies and drought severity, defined by anomalies in vapor pressure deficit (VPD) and soil moisture. The findings reveal significant drought effects in 2022, distinguishing this year from other extreme years, such as 2003 and 2018. This timely research contributes to our understanding of the drought-ecosystem nexus under intensifying climate extremes. It aligns closely with the theme of this special issue and likely sparks considerable interest within the drought research community.*

*My primary concern revolves around the manuscript's conclusion on the legacy effects and declining forest resilience given the higher damage found in 2022, which are stressed across the manuscript (e.g., lines 38-40, lines 319-323, lines 395 - 399). The presented results may not be straightforward enough to support these claims, and could potentially be misleading. For instance, the extensive forest damage highlighted in e.g., Fig. 3, along with changes in vegetation-climate relationships shown in Fig. 7 and 8 might simply reflect the varied geographical locations of droughts and heatwaves over the years (i.e., 2003,2018, and 2022). Notably, the broader impact areas in 2022, predominantly in southern Europe—a region already limited by water availability in normal years (Fig. 2)—suggest that ecosystems there may have a smaller drought-tolerance margin and are more sensitive to droughts compared to Northern Europe forests. This may explain the smaller overall negative effects on forests in 2018 even under a larger relative drought, because the drought may not hit the "right" vulnerable target. The observed variations likely arise from examining distinctly different ecosystems (e.g., evergreen needleleaf forests versus deciduous broadleaf forests, as shown in Figure 1), rather than a general increase in ecosystem vulnerability or a decrease in resilience due to legacy effects. Therefore, I recommend tempering the conclusions. Simply summarizing the main findings would provide ample insight without overextending the implications.*

Response: Thank you for your feedback. In the revised version of the manuscript, we will tone down the conclusion and summarize the main findings to avoid overextending the implications of our findings. See our response to further comments (also from reviewer Nr. 2) please to see how we plan to do this.

*Other minor comments and questions:*

*Ln 85-87: "Elevated temperatures can …. reducing anthropogenic CO2 emission": I can not see how this could make anthropogenic CO2 reduce, further clarification is needed.*

Response:
We will rewrite this sentence to make it clear. We will write:
"Elevated temperatures can also increase rates of respiration from the soil and from the trees which leads to reduced net capacity of forests for carbon uptake and capacity of forests for reducing anthropogenic $CO_2$ emissions"

*Ln 130: For analysing the total areas of affected forests, would be good to mention in the Methods section how grid datasets at degree resolution could be converted to surface area in km2, if trigonometric difference in the sphere is considered.*

Response:
The trigonometric difference arising due to Earth's spherical shape (Ellipsoid) was considered while converting the area from grid datasets to $km^2$. We will add the details in the methods section.

*Ln 147: the use of NIRv is quite interesting. But EVI could also correct canopy background noise and should be less sensitive to soil conditions. What is the advantage of using NIRv over EVI? Perhaps more information is needed.*

Response:
Studies that did a comparison of NIRv and EVI against flux-estimated GPP show that NIRv reflect vegetation phenology better than EVI (see Zhang et al. 2022). We will clarify this in the methods section.

*Ln 236: Would be good to indicate what the shaded areas mean for Fig. 3 and other similar figures (i.e., the orange, white and green). I notice that you have labelled them in Fig. 5, but why not do that earlier?*

Response:
We will add to the caption of the figure: "Orange-shaded area marks below normal and green-shaded area marks above normal conditions.

---

## Author Response (AR1)

**Discussion of "Effect of the 2022 summer drought across forest types in Europe"**

Reviewers' comments are marked in black and authors´ responses are marked in blue.

Gharun et al.

6 August 2024

We thank the editor and the referees for their valuable comments and insights, which have greatly contributed to enhancing the manuscript. Below, we first respond to the handling editor´s comments and then outline all the modifications made in the revised version of our manuscript. Furthermore, we provide a concise summary of the detailed responses provided to the referees during the initial stage of the revision process.
We understand that the main critiques on our paper were regarding the 1) clarity of the methods and descriptions of results and 2) conclusions that can be drawn from our analysis (e.g., with respect to any legacy effects). We have addressed these points and also addressed all the line-by-line comments of the Referee 1 and 2. Finally we made changes to improve the overall clarity of the manuscript.

**Response to handling editor comment**

Handling editor´s comment:

The manuscript has been evaluated by two reviewers. Both reviewers consider this work interesting and suitable for the special issue. However, they raise some concerns regarding the clarity of the methods, descriptions of results and conclusions that can be drawn. They provide a number of suggestions to improve the manuscripts. I agree on this evaluation. Of particular importance is to avoid drawing conclusions relative to legacy effects if these cannot be proved beyond doubt based on the analyses presented.
Moreover, it is important that the effects of implicit assumptions or consequences of chosen data and methods are thoroughly discussed. An example in this sense is the soil depth considered: 0 to 7 cm is clearly too shallow even as depth in which most of the roots are located in forests (see e.g. Bachofen et al New Phytologist 2024 https://doi.org/10.1111/nph.19762). That conclusions do not depend on such choice needs to be tested or, at the very least, discussed.
From my own reading, I would add that the focus on June-July-August over the entire Europe needs justification: while these months fall under the growing season in all locations, they represent a different proportion of the entire growing season depending on the location (see e.g. Wu et al 2022 Geophysical Research Letters https://doi.org/10.1029/2022GL098700). The implications of the focus on June-July August need to be discussed. I also note that transparent meaning and use of the terminology is important, particularly in the case of drought, which has complex and variable definitions. The authors may even consider to run a sensitivity analysis on how they define drought based on the z-score, to check the robustness of their results.
Note: the comments contain several references, mostly in support of the reasoning there. Do not feel pressured to add any reference you do not deem necessary.

Response:
Thank you for your feedback and the opportunity to revise our manuscript. We have revised the discussion of our results to avoid drawing conclusions about the legacy effects, since this was a speculation that could not be quantified from our analysis. Regarding the justification for selecting the dataset considered (soil depth at 0-7 cm) we ran additional tests, and show that our results remained unchanged. We address this point in full detail in response to the comments of one of the reviewers.
We restricted our analysis to the months of June-July-August as the focus of our paper as it can be seen from the title, was on the summer months. The ecophysiological growing season, as the editor puts it correctly, deviates from the calendar summer, depending on the region. But also depending on the observation method - for a detailed discussion on this topic we refer you to a recent review by Körner et al. 2023 "Four ways to define the growing season" (Ecology Letters https://doi.org/10.1111/ele.14260). Here our objective was not to restrict the analysis to the growing season, but we selected these months so our study is 1) comparable with existing studies focused on the summer drought 2) to capture the peak of the warm and dry conditions across Europe, that would be most stressful for the vegetation functioning, from the perspective of heat and water supply. We have clarified this justification in the Methods section "Drought detection and statistical data analysis".
Regarding the definition of drought as presented in our paper, we have added an extra section to the Methods to justify the robustness of our results (see below for more detail please).

**Reviewer #1**

This study explores the impacts of the 2022 summer droughts on the forest canopy across different European land covers, utilizing two remote-sensing vegetation datasets: near-infrared reflectance of vegetation (NIRv) and solar-induced fluorescence (SIF). The authors analyzed the extent of forests affected and examined the correlations between canopy condition anomalies and drought severity, defined by anomalies in vapor pressure deficit (VPD) and soil moisture. The findings reveal significant drought effects in 2022, distinguishing this year from other extreme years, such as 2003 and 2018. This timely research contributes to our understanding of the drought-ecosystem nexus under intensifying climate extremes. It aligns closely with the theme of this special issue and likely sparks considerable interest within the drought research community.

My primary concern revolves around the manuscript's conclusion on the legacy effects and declining forest resilience given the higher damage found in 2022, which are stressed across the manuscript (e.g., lines 38-40, lines 319-323, lines 395 - 399). The presented results may not be straightforward enough to support these claims, and could potentially be misleading. For instance, the extensive forest damage highlighted in e.g., Fig. 3, along with changes in vegetation-climate relationships shown in Fig. 7 and 8 might simply reflect the varied geographical locations of droughts and heatwaves over the years (i.e., 2003,2018, and 2022). Notably, the broader impact areas in 2022, predominantly in southern Europe—a region already limited by water availability in normal years (Fig. 2)—suggest that ecosystems there may have a

smaller drought-tolerance margin and are more sensitive to droughts compared to Northern Europe forests. This may explain the smaller overall negative effects on forests in 2018 even under a larger relative drought, because the drought may not hit the "right" vulnerable target. The observed variations likely arise from examining distinctly different ecosystems (e.g., evergreen needleleaf forests versus deciduous broadleaf forests, as shown in Figure 1), rather than a general increase in ecosystem vulnerability or a decrease in resilience due to legacy effects. Therefore, I recommend tempering the conclusions. Simply summarizing the main findings would provide ample insight without overextending the implications.

Response:
Thank you for your feedback. We have removed the statements related to a potential legacy effect (lines 38-40, lines 319-323, lines 395 - 399) to avoid any misleading statements. In the revised version of the manuscript we have toned down the conclusion and summarized the main findings to avoid overextending the implications of our findings. We mention that the canopy damage that we detected shows an evident vulnerability of the forests to these extreme conditions. For more detail please see our response to further comments (also from reviewer Nr. 2).

Other minor comments and questions:

Ln 85-87: "Elevated temperatures can …. reducing anthropogenic CO2 emission": I can not see how this could make anthropogenic CO2 reduce, further clarification is needed.

Response:
We have rewritten this sentence to make it clear. The statement now reads:
"*Elevated temperatures can also increase rates of respiration from the soil and from the trees which leads to reduced net capacity of forests for carbon uptake and capacity of forests for reducing anthropogenic $CO_2$ emissions*"

Ln 130: For analysing the total areas of affected forests, would be good to mention in the Methods section how grid datasets at degree resolution could be converted to surface area in km2, if trigonometric difference in the sphere is considered.

Response:
The trigonometric difference arising due to Earth's spherical shape (Ellipsoid) was considered while converting the area from grid datasets to $km^2$. We added the details in the Methods section "Land cover dataset" as following:
*"Area of each grid cell was calculated using trigonometric equations considering the latitudinal and longitudinal variations arising due to Earth's spherical shape (Ellipsoid)."*

Ln 147: the use of NIRv is quite interesting. But EVI could also correct canopy background noise and should be less sensitive to soil conditions. What is the advantage of using NIRv over EVI? Perhaps more information is needed.

Response:
Studies that did a comparison of NIRv and EVI against flux-estimated GPP show that NIRv reflects vegetation phenology better than EVI (see for example Zhang et al. 2022). We have clarified this in the Methods section "Forest canopy response dataset".

Ln 236: Would be good to indicate what the shaded areas mean for Fig. 3 and other similar figures (i.e., the orange, white and green). I notice that you have labelled them in Fig. 5, but why not do that earlier?

Response:
We have added to the caption of the figure: "*Orange-shaded area marks below normal and green-shaded area marks above normal conditions*".

**Reviewer #2**
the paper analyzes the relationship between two climate variables relating to drought (soil moisture & VPD) and the productivity of forests as measured by the proxies NIRv & SIF, quantifying the severity of drought and the productivity reaction through the z-score. A visual comparison is provided between different drought periods in terms of the z-score of the productivity metrics and the z-score of the meteorological metrics. The relationship of these responses is then related to individual forest types and the response strength compared in terms of partial correlations.

The paper addresses some relevant scientific questions in terms of the scale and impacts of a specific recent drought and poses questions about why there may be a different response compared to two previous droughts. It is good that it uses multiple novel metrics of vegetation productivity in both NIRv and SIF. It is relevant to the current special edition and has some novelty, in that it analyses the impact of the 2022 heatwave on vegetation. However, in its current form it needs a lot more detail and clarity to support some of its conclusions. A deeper explanation of the analysis and some statistical clarifications, alongside extra supporting analysis to support the results (such as further drought indicators and more than just the 0-7cm of topsoil), perhaps reconsidering which areas considered in the analysis are actually drought affected. A thorough proofreading before re-submission is important. A toning-down of some of the findings/conclusions is also required and would be helped by a wider consideration of what may be contributing to the observations.

Response:
Thank you for your feedback. In the revised manuscript we have improved the clarity of the paper and provided a deeper explanation of the analysis, together with additional analysis to support our results. We have also toned down some of the conclusions to avoid overextending the implications of our findings. For more detail please see our response to each specific comment.

Specific comments

Clearer introduction and explanation of the figures
The paper would benefit from a clearer explanation of some of the figures, and a separation between sentences explaining/defining what the figures are and sentences describing the results in the figures. Clearer captions would also help this: figure captions should be standalone, i.e., descriptive enough to be understood without having to refer to the main text. A good positive example is figure 2 which describes clearly what the content is with a clear line introducing the figure by description, before later describing the results.
L215 : 'Figure 2 shows the extent and magnitude of anomalies (z-score) of VPD and top layer (0-7 cm) soil moisture content during the summer months in 2003, 2018, and 2022 across the entire region of Europe.' The caption is also reasonably descriptive. However, in figure 3 there is no similar sentence introducing the figure. Instead it is referenced in brackets without explanation/description:

Response:
We have added explanations to separate sentences explaining/defining the figures and sentences describing the results in the figures. For Figure 3 we have added "*Figure 3 shows the intensity of atmospheric and soil drought via z-score values of VPD and SM anomalies over the summer months (JJA) in 2003, 2018, and 2022. The total affected area displayed in Figure 3 is the sum of all pixels within the given z-score bin during the summer period where z-scores are averaged for each bin for the summer period.*"

L226: 'Restricted to forested areas, atmospheric and soil drought was 55% and 58% more extensive in 2018 compared to 2022 (and both years more extensive than in 2003, Figure 3).' This doesn't give us information on important facts of the figure: Are these the z-scores for the entire JJA period (like in the previous figure – compared to the previous JJA periods) or is it for 8-day periods? I would assume the former, but then the NIRv and SIF is described only in terms of 8-day, so it's not clear if it has been averaged or aggregated. Is the forest area affect the total summation of all pixels with the given binning of z-score during that summer period? Is the z-score the average for the full JJA for each pixel? All this useful information is missing, and even more could be added to help the reader understand.

Response:
The z-score values are aggregated for the entire summer period (JJA). To clarify the statement, we have added "*The total affected area displayed in Figure 3 is the sum of all pixels within the given z-score bin during the summer period where z-scores are averaged for each bin for the summer period.*"

The same is true for figure 4, figure 5, figure 6, 7, 8, & 9. It would be nice to have a specific reference to each of them in the text describing first what we are seeing (i.e. what is the figure of) before describing the results (i.e. what does it tell us). Particularly as we are missing information on the time periods described (8-day or summer) and – especially for 7/8 - whether we are mixing space and time pixels (i.e. is each point a particular pixel in space and a particular time period – this is very important for the interpretation).

Response:
We have added more information about each figure to the text, to describe them first.
For Figure 4 we added: "*The intensity of GOSIF and NIRv anomalies over the summer months (JJA) in 2003, 2018, and 2022 are displayed in Figure 4. The extent shown in Figure 4 is the sum of all pixels within the given z-score bin during the summer period (z-scores are averaged for each bin).*"
For Figure 5 we added: "*Figure 5a shows the GOSIF anomalies (z-score) across all forested areas in Europe. The intensity and extent of the GOSIF anomalies during the summer months (JJA) in each year are shown for different forest types in Figure 5b.*"
For Figure 6 we added: "*Figure 6a shows the anomalies of NIRv (average z-score over the summer months) across all forested areas in Europe. The intensity and extent of the NIRv anomalies during the summer months (JJA) in each year are shown for different forest types in Figure 6b.*"
For Figure 7 and 8 we added: "*Figure 7 shows the spatial regression between standardized SIF anomalies with (a) VPD and (b) SM and Figure 8 shows the spatial regression between standardized NIRv anomalies with (a) VPD and (b) SM over the drought areas in summers 2003, 2018 and 2022.*"
For Figure 9 we added: "*Figure 9 shows the temporal partial correlation coefficient of GOSIF with SM and VPD during summer months (JJA) for areas identified as affected (Figure 9a) and not affected (Figure 9b) by drought.*"
In addition, we added a short statement to all figure captions that the z-scores are derived for the summer months (JJA).

For Figure 7:
L273: 'With the increase in VPD positive anomalies (i.e., increased atmospheric dryness), SIF values declined across all forest types, across all years, except in 2022 in the WSA, and in 2018 and 2022 in EBFs (Figure 7).' However, the caption of figure 7 does not mention any VPD anomalies (it seems to have the actual VPD and SM – with units). So which is it? Also what are each of the points? Are they individual pixels? And if so are they at 8-day values or JJA?

Response:
We have corrected the statement and written instead: "*With the increase in VPD (i.e., increased atmospheric dryness) GOSIF values declined across all forest types, across all years, except in 2022 in the WSA, and in 2018 and 2022 in EBFs*".
We have clarified it in the caption of Figure 7 that each of the points are the values for JJA.

Similarly for figure 9: L292 'The SM and VPD anomalies across all forest types correlated well, but across DBFs the dryness in the atmosphere and the dryness in the soil were most correlated (Figure 9).' In this description we are talking about SM and VPD anomalies, but in the caption, there is no mention of the anomalies. So are we talking about anomalies or not?

Response:

The temporal partial correlations shown in Figure 9 are related to the absolute values and not anomalies. We corrected this statement and wrote: "*The SM and VPD values across all forest types correlated well, but across DBFs the dryness in the atmosphere and the dryness in the soil were most correlated (Figure 9)*".

The paper needs more precise language and a clear explanation of where the data in the figures comes from and what the figure contains. This should be separate from trying to describe the patterns. As it stands it can be confusing and a struggle to read, and therefore difficult to understand and judge the scientific content as well as difficult to reproduce (especially as code and data is not released). I expand on this in the minor comments, but it isn't fully clear what is being done in some of the figures. A slower more explicit explanation of each figure is necessary, including more information in both the text and the caption.

Response:
We have followed the reviewer´s recommendation and clarified the explanation of the data and what each figure contains. We have added a more explicit explanation of each figure in the text and in the figure captions (see responses below for more detail).

Clarification of the SIF data used

The paper would benefit from more discussion of both what SIF actually is and the SIF product used, including how it was developed. Indeed there is not much explanation of how the SIF signal originates and therefore why it might be affected by drought, other than SIF is a proxy for photosynthesis and 'provides information about the physiological response of forest photosynthesis' (a more complete description of SIF in the introduction would be nice).

Response:
Thank you for the suggestion. We refrained from writing a complete description of SIF in the introduction as the paper is about drought impact on forest and not SIF (although SIF is a tool measuring the impact). However, we have improved our description of SIF in the Methods (subsection: 'Forest canopy response dataset') and added its implications on our results in the Discussion section of the manuscript.

The improved description in the method section is as following:

*"In order to assess the forest canopy response to drought stress, we used two satellite-based proxies: 1) The structure-based NIRv (near-infrared of vegetation index derived from MODIS (Moderate Resolution Imaging Spectroradiometer; 8-day 500m x 500m MOD09Q1 v6.1 product) which is calculated using surface spectral reflectance at near-infrared band ($R_{NIR}$) and red band ($R_{Red}$) as shown in Equation 2 (Badgley et al. 2017). The calculated NIRv at 500m resolution was aggregated (averaged) at a 0.05°✕0.05° resolution.*

$$NIR_V \ = \ R_{NIR} \times \frac{R_{NIR} - R_{Red}}{R_{NIR} + R_{Red}} \qquad\qquad (2)$$

*2) The physiological-based reconstructed global OCO-2 (Observation Carbon Observatory - 2) solar induced fluorescence (GOSIF) dataset. Solar-induced fluorescence (SIF) is an energy flux (unit: $Wm^{-2}\mu m.sr^{-1}$) reemitted as fluorescence by the chlorophyll a molecules in the plants during photosynthesis (Baker, 2008). Recent extensive research studies have linked SIF to vegetation photosynthesis, confirming it as a proxy for the ecosystem's gross primary productivity (GPP) (Li et al. 2018; Magney et al. 2019; Shekhar et al., 2022). The GOSIF dataset was created by training a Cubist Regression Tree model to gap-fill SIF retrievals from OCO-2 satellite. This was done using MODIS Enhanced Vegetation Index (EVI) and meteorological reanalysis data from MERRA-2 (Modern-Era Retrospective analysis for Research and Applications), which includes photosynthetically active radiation (PAR), vapor pressure deficit (VPD), and air temperature (see Li and Xiao, 2019). We downloaded GOSIF data set (v2) from the Global Ecology Data Repository (http://data.globalecology.unh.edu/data/GOSIF_v2/, last accessed 25 July 2024). The GOSIF was available from 2000-2022 at 8-day temporal scale with a spatial resolution of 0.05°✕0.05° (Li and Xiao, 2019).*

*GOSIF signals provide information about physiological response of forest photosynthesis while NIRv (a recently developed vegetation index) signals provide information about the health status of the canopy."*

The modified parts of discussion section reads as following:

*"SIF dataset used in this study, i.e., GOSIF product, has proved to be a reliable proxy for vegetation gross productivity, as comparison with ground-based flux measurements show (Shekhar et al. 2022; Pickering et al. 2022). It should be noted that the GOSIF data are estimated using a machine learning model that was trained with OCO-2 SIF observations, MODIS EVI data, and meteorological reanalysis data. Therefore, the meteorological data that is used in our analyses are not completely independent of our SIF data. However, this does not affect our findings. A recent study used both GOSIF and original OCO-2 data to examine the impacts of the 2018 U.S. drought on ecosystem productivity and found that SIF based on these two datasets had similar responses to drought (see Li et al. 2020)."*

It also isn't made clear exactly what SIF product is being used. In the abstract they say that 'the OCO-2 solar induced fluorescence' dataset, whilst reading deeper it appears it is the GOSIF product (indeed on figure 5, the label GOSIF is on the x-axis, a label which is not introduced/explained anywhere else in the text). If it is the GOSIF dataset, which it appears from the reference Li, X., Xiao, J. (2019), this should be explained more clearly and there should be some discussion of its derivation and its limitations.

Response:
As responded in response to your query before, in the revised version of the manuscript we have 1) clarified in the Methods section what SIF is, 2) clarified what product we have used (GOSIF product) and explained its reconstruction process, and 3) replaced "SIF" with "GOSIF" everywhere throughout the paper where the focus is the product that we have used.

This is important because the reader is given the impression that 'SIF was available 2000-2022 at 8-day temporal scale with spatial resolution of 0.05x0.05' from the 'OCO-2 SIF dataset', when in reality no satellite dataset exists. OCO-2 was only launched in 2014 and has discontinuous spatial coverage. GOSIF is a product derived from OCO-2 data and combined with remote sensing data from MODIS, and meteorological reanalysis, using a data-driven approach to predict the SIF in the earlier years. The paper should make this distinction clear. Whilst it is mentioned in the methods 'we used two satellite-based proxies: 1) ... 2) the physiological-based reconstructed global OCO-2 solar induced fluorescence (SIF). ' the word 'reconstructed' really skips over details which are highly relevant to the paper results. As one of the main conclusions of the paper is that there is a difference in the SIF response between 2003 and 2022, and that the difference is due to meteorological drivers, it is important to note that the SIF signal in 2003 is a modelled/predicted signal, modelled/predicted based on those same meteorological drivers, whilst the SIF response in 2022 is a measured OCO-2 signal (although also somewhat modelled due to gap filling). Therefore there should be some discussion of this limitation of the study – we are not comparing like with like, and the modelling may have a confounding effect (i.e. we are looking at the impact of our 'independent variable' meteorology on our 'observed variable', SIF, but our independent variable is also used to predict our 'observed' variable, so our 'independent' variables are not really independent of SIF). I don't think this is enough to cause a fundamental problem for the results, but it deserves discussion.

Response:
In fact, SIF in both 2003 and 2022 was modeled using the predictive SIF that we trained with original, OCO-2 SIF soundings, EVI, and meteorological data. So we are actually comparing like with like. We have briefly discussed the confounding effect the reviewer pointed out in the discussion as following: "*It should be noted that the SIF data used in this study (i.e., GOSIF) are estimated using a machine learning model that was trained with OCO-2 SIF observations, MODIS EVI data, and meteorological reanalysis data. Therefore, the meteorological data that is used in our analyses are not completely independent of our SIF data. However, this is not likely to affect our findings. A recent study used both GOSIF and original OCO-2 data to examine the impacts of the 2018 U.S. drought on ecosystem productivity and found that SIF based on these two datasets had similar responses to drought (see Li et al. 2020).*"

What is a drought and what is considered stress

One problem with the paper is the definition of what is a drought . The word drought is used extensively in the publication, including in the title, as well as being specifically defined in the paper, and referenced throughout the paper. My understanding is that there are a few ways meteorologists and vegetation specialists may define it, such as:
1) 'A drought is a period of unusually persistent dry weather that continues long enough to cause serious problems such as crop damage and/or water supply shortages… low precipitation over an extended period of time' NASA https://gpm.nasa.gov/resources/faq/what-drought-and-what-causes-it

2) drought is defined as a deficiency of precipitation over an extended period of time (usually a season or more), resulting in a water shortage NOAA https://www.drought.gov/what-is-drought/drought-basics

3) "A period of abnormally dry weather sufficiently long enough to cause a serious hydrological imbalance." American Meteorological Society

4) regarding specifically the 2022 drought: https://climate.copernicus.eu/esotc/2022/drought it was not just hot/dry conditions during the summer, 'A persistent lack of precipitation was observed from winter 2021/22 onwards and, for the year as a whole, surface soil moisture was the second lowest in the last 50 years. Higher-than-average temperatures and a sequence of heatwaves that started in spring and continued throughout summer sustained and enhanced drier-than-average conditions.'

Whilst there is no specific quantifiable criteria to define a drought (it is an area subject to debate), it is clear that it relates to persistent dry conditions enough to have a serious impact. As there is this open discussion on what defines a drought, publications about drought (as opposed to say – water stress), might often use a couple of different metrics.

In the paper, drought is defined as

'Areas were categorized as under drought if VPDz > 1 & SMz < -1, and as normal areas if -1 < VPDz < 1 & -1 < SMz < 1.'

i.e. based on 1sigma deviations of Vapour pressure deficit and soil moisture (calculated across the JJA summer period from 8-day SM&VPD data in the 0-7cm layer) from their long-term means. There is no reference given to literature to support this definition of drought. Is there some literature to support this? At the least, the paper would benefit from a larger discussion on this open question of what defines a drought and why this definition is appropriate.

Response:
We agree with the reviewer that the term "drought" is frequently used in publications and is often not specifically defined in ecological studies. For a comprehensive review of this issue, we refer to Slette et al. (2019), "How ecologists define drought, and why we should do better (Global Change Biology 10.1111/gcb.14747). Most of the drought definitions mentioned by the reviewer are largely based on precipitation data and are commonly used for hydrological studies. We agree that it is important to have a clear definition of conditions that are extremely dry, based on indicators that are ecologically relevant. As the reviewer also mentions, we have defined drought as persistent dry conditions enough to have a serious impact on the ecosystem. Regarding the choice of the variables, we know from the body of literature that both SM and VPD directly influence vegetation functioning and thus are suitable proxies for identifying environmental limitations to plant physiological functioning.
Regarding the definition and identification of drought: we have defined drought as conditions of when the soil moisture and VPD are below normal and above normal respectively, i.e., presence

of soil and atmospheric dryness. In drought identification studies, classification of 'normal' (not to be confused with normal distribution), 'drought' (used synonymously with 'dry'), or 'wet', is largely done using a standardized index, such as SPI (Standardized Precipitation Index), SPEI (Standardized Precipitation Evapotranspiration Index), Z-score among others, as highlighted in the review 'A review of drought concepts' by Mishra and Singh (2011). All studies that use a standardized index for classification, classify "normal" conditions when the index is between -1 and 1, and "drought" conditions when the index is < -1, and "wet" conditions when the index > 1. See for example Table 2 of Jain et al. (2015), Table 4 of Wable et al. (2019), Table 2 of Dogan et al. (2012), and Table 1 of  Tsakiris and Vangelis (2004).

In this study, we classify drought conditions using standardized metrics for soil moisture (SM) and vapor pressure deficit (VPD), specifically their z-scores. This approach aligns with the established drought identification methods in the literature. Such classification of 'normal' (and thus, 'above normal' and 'below normal' used in this study) based on z-score (also called standardized anomalies) can be done for any meteorological and/or response variables such as NIRv and GOSIF done in this study, making the narration of results coherent across different variables.

In the revised manuscript, we have incorporated the above-mentioned explanation into the Methods section to clarify our drought definition and identification approach as following:

*"In drought identification studies, classification of 'normal' (not to be confused with normal distribution), 'drought' (used synonymously with 'dry'), or 'wet', is largely done using a standardized index, such as SPI (Standardized Precipitation Index), SPEI (Standardized Precipitation Evapotranspiration Index), Z-score among others (see Mishra and Singh, 2011). All studies that use a standardized index for classification, classify "normal" conditions when the index is between -1 and 1, and "below normal" conditions when the index is < -1, and "above normal" conditions when the index > 1 (Jain et al., 2015, Wable et al., 2019, Dogan et al., 2012, Tsakiris and Vangelis, 2004). In this study, we classify drought conditions when the soil moisture is below normal (SMz < -1) and VPD is above normal (VPDz > 1), i.e., presence of both soil dryness AND atmospheric dryness. This approach (threshold-based using standardized anomalies) aligns with established drought identification methods in the literature and is relevant for studying drought impact on forests as we know from the body of literature that both SM and VPD directly influence vegetation functioning and thus are suitable proxies for identifying environmental limitations to plant physiological functioning. Furthermore, such classification of 'normal' (and thus, 'above normal' and 'below normal' used in this study) based on z-score (also called standardized anomalies) can be done for any meteorological and/or response variables (such as NIRv and GOSIF done in this study), making the narration of results coherent across different variables."*

It is not entirely clear to me if some of the figures are showing 8-day VPD and SM in the data, or if the JJA anomaly is the only one used. If it is only the JJA anomaly used then I think it might just fit the description of 'extensive/persistant' (though it would be important to add more of a discussion including literature on what defines a drought and the fact that period preceeding the summer may play a role). However if some of the figures are showing the 8-day anomaly of SM and VPD then I'm not sure we can use the word drought over such a short time period (sure it

might be a period of low VPD or soil moisture during a drought, but the fluctuation itself is not defining a drought).

Response:
The JJA anomaly is used. We have clarified this throughout the text.

Whilst these VPD/SM fluctuations do of course relate to drought, drought is an extensive and meaningful extreme climate event, and. I'm not sure if many people would define a 1s.d. downward fluctuation in VPD and SM in the topsoil layer as 'under drought'. Particularly if we are looking at the 8-day VPD and SM (which is not 'an extended' or 'persistant' period of time in the growing season of a tree. Trees have roots that go far beyond the first 7cm (https://bg.copernicus.org/articles/17/5787/2020/), and so a slightly dry period affecting the top layer might not be enough to cause damage or significant browning. Forests are generally sufficiently adapted to tolerate (1sd) temporary fluctuations in soil moisture and air dryness.

Alternatively, if we are taking the 2003, 2018, 2022 summers as established droughts (given in the literature), and then within these drought categories we are looking at further water stress variability within the drought affected region, then we could drop the definition of drought and keep the analysis (i.e. we are using a definition of drought defined by other literature sources). However it wouldn't make so much sense in that case to include areas that are not part of the drought affected regions (e.g. Spain in 2018, or Northern Europe in 2022). Personally I would define < -1 s.d. SM and > 1 s.d. VPD as under a some water stress. With > 2/2.5/3 s.d. perhaps we can maybe start thinking about a more major (or 'significant') water stress and drought (if persistent) – this is similar to definitions here: https://www.science.org/doi/10.1126/sciadv.aba2724#sec-5 .

Response:
The years that we have selected were generally identified as extreme years as unprecedented conditions of high air temperature and low precipitation levels were recorded across many regions in Europe. For our analysis we thought it was necessary to provide a clear definition of the conditions we consider stressful for the vegetation in terms of limiting water availability in the soil and in the atmosphere, and apply the same definition for all regions (regardless if the trees in a certain region have the mechanisms to cope better with the drier conditions). Without a clear metric it would be subjective to keep certain regions out from the analysis. We chose a "relative" metric (as opposed to a fixed threshold) and assessed water availability in all regions relative to the conditions of that particular region.
Regarding the definition for a major (or "significant") stress, we have this classification at the higher z-score levels. The -1 < z-score < 1 range specified the "normal" conditions and outside this we have different degrees of water stress depending on the high z-scores. Please see our response to your previous query, for more clarification.

Including layers beyond the 0-7cm range would benefit the analysis as deeper droughts are more stressful to the vegetation with deep roots (see

https://nhess.copernicus.org/articles/23/1921/2023/ where 1m soil is considered or https://royalsocietypublishing.org/doi/10.1098/rstb.2019.0507 where 2.89m soil is considered).

Response:
Yes, we agree that drought in the deeper soil layers can be more stressful for the vegetation that has a deeper rooting system. However, based on our previous experience, drought propagates similarly across different depths (see Figure 4 of Lal et al., 2023). Therefore, we assumed that the drought area detected using the 0-7cm SM data will be similar to the drought area detected based on 0-100cm SM data.

In order to confirm this and address the reviewer's query, we now compared areas detected as affected by drought based on the dataset from the top 0-7 cm soil layer, with drought-affected areas detected based on the soil moisture data within top 0-100 cm. Our comparison showed that the difference between the area detected under drought based on 0-7 cm SM and 0-100 cm SM is less than 5% as indicated in figures below. Furthermore, the standardized SM anomalies of 0-7 cm and 0-100 cm were highly correlated (r = 0.95), thereby not affecting any of our sensitivity results. Thus, we remain with using the 0-7 cm of SM data for this study.

[Figure]

[Figure]

Figure: Comparisons of standardized SM anomalies (SMz) of 0-100 cm SM layer (y-axis) and 0-7 cm SM layer (x-axis) during the three drought years of 2003, 2018, and 2022 in Europe across forested areas, in terms of spatial coverage (top plot), percentage of total area affected (middle plot) and correlation of the anomalies (bottom plot).

We have added this comparison to the supplemnetary material.

If we are looking at 8-day VPD and SM anomalies (I don't think we are, but I say just in case, as it is not clear), however, then we are not really fitting the definition of a 'persistent' time period. In order to use the word drought, and compare the extremity of droughts, I think it would be necessary to use a specific drought index, such as Palmer Drought Severity Index (PDSI) (Palmer, 1965), Standardized Precipitation Index (SPI; McKee et al., 1993), and maybe better SPEI https://spei.csic.es/ . I think this would benefit the analysis even if we are not considering the 8-day anomalies. Such a predefined index could be used in conjunction with the current work and fits nicely as another metric showing water stress relative to the average. It would satisfy the requirement of a persistent time period (as the analysis could consider both 3-month and 6-month periods). It would hopefully not require too much extra work (the downloading of precipitation – already present it seems - and evapotranspiration datasets and calculation of the index values using one of many available packages) or many more figures (a third panel for fig3 fig7 and fig8).

Response:
No, we are not looking at the 8-day VPD and SM anomalies. We are looking at the average anomalies for the summer months (JJA). Water availability (or the lack of it) can be identified with three methods: 1) fixed thresholds (e.g., of precipitation amount, soil water potential, etc.) 2) relative thresholds (e.g., z-score and anomaly of climate drivers) and 3) drought metrics.
We chose the second approach and quantified a relative threshold, and applied it to atmospheric and soil moisture data since we believe that VPD and SM incorporate the feedback from the vegetation to the climate conditions (mediated for example by the actual evapotranspiration) whereas a drought index such as SPEI reflects only the potential for evapotranspiration and not the feedback from the vegetation response.

The area of drought considered

Additionally, it seems we are considering the same time period JJA and the full extent of Europe for each of these 'droughts'. But the heatwaves are not directly comparable because there are different areas of focus of the drought in each separate drought event, and the droughts lasted different time spans. So in the end we aren't really comparing similar data between years – we are making large spatial and temporal averages (different European summers) and applying them to specific spatially and temporally constrained events (heatwaves). In particular it seems we are including non-drought pixels (e.g. Spain/Italy in 2018) in the comparison of between the different drought years and so drawing conclusions based also on the reaction of non-drought areas. This is most clear in the following example: Scandinavia was not really affected by the 2022 heatwave as your figure shows (as well as https://www.ecmwf.int/en/about/media-centre/news/2022/european-heatwaves-june-2022 ) yet this region is included in the analysis. One of the conclusions is: 354: 'While deciduous broad-leaved forests were most negatively affected by the extreme conditions in 2022, Evergreen Needle-Leaf Forests (ENF) distributed in

northern regions of Europe showed enhanced canopy greening and SIF signals, through benefiting from the episodic warming.'

Which is explained with an extensive discussion on physiology:

L358: 'The mechanisms to cope with the level of drought stress, vary largely among forest types, and depend on a combination of characteristics that control water loss through the coordination of stomatal regulation, hydraulic architecture, and root characteristics…'

Another, simpler, way to interpret this would be that ENF is mostly present in Scandinavia. Scandinavia was not really affected by the 2022 heatwave. Therefore Europe's ENF forests were not really affected by the heatwave. Is this mechanism accounted for in the analysis? I don't think it is wrong to say L358, however I'm not sure the methodology of the paper is sufficient to support the conclusion that the difference in response was due to physiology, as opposed to a given region just not being involved in that particular heatwave. This conclusion of the paper may risk suggesting that ENF forests are more drought safe than other forests – which is probably not being demonstrated in the analysis.

Response:
We agree that our wording here implies a wrong interpretation. We changed this statement in the revised manuscript and wrote in the Discussion:
"*While deciduous broad leaved forests were negatively affected by the extreme conditions in 2022, Evergreen Needle-Leaf Forests (ENF) distributed in northern regions of Europe were not exposed to extremely dry conditions in 2022 and even showed enhanced canopy greening and GOSIF signals, through benefiting from the episodic warming (Forzieri et al. 2022). Under similar drought conditions, the mechanisms to cope with the level of drought stress vary largely among forest types,..*"

Legacy effect

Another key conclusion drawn is that the 'higher degree of canopy damage in 2022' suggests 'declining resilience of forests to drought' and 'points to a legacy effect'. However by combining different spatial areas and mixing different forests I'm not sure we are really testing a 'legacy effect'. To test a legacy effect or declining resilience we should really control for the impact of droughts on the same forest areas across compounding years and observe if similar sized perturbations cause worsening compounding effects. Taking Europe-wide spatio-temporal averages of vegetation response (across various distributed forest types) between two very different averaged periods of 'drought' (i.e. 1sd fluction in 0-7cm topsoil water and VPD) and saying that the second period was more severe despite lower 'drought', because of a legacy effect and therefore shows a decline in resilience is a very strong conclusion from a very broad methodology. Assuming I have understood the methodology correctly for this conclusion – feel free to point out if I have missed something. Additionally, it seems clear that the 2022 drought, whilst perhaps less extreme during the summer in question was coupled with a winter drought over a prolonged period. Described here: ' https://climate.copernicus.eu/esotc/2022/drought 'A persistent lack of precipitation was observed from winter 2021/22 onwards and, for the year as a whole, surface soil moisture was the second lowest in the last 50 years. Higher-than-average

temperatures and a sequence of heatwaves that started in spring and continued throughout summer sustained and enhanced drier-than-average conditions.'

Additionally the preconditions of the 2018 drought (spring/winter) were less extreme: https://www.science.org/doi/10.1126/sciadv.aba2724 How do we know from the analysis that the 'higher degree of damage in 2022' compared to 2018 was not due to a prolonged drought in 2022 beyond the period considered (ie winter), even if there was slightly more water available during the summer months?

Potentially this can be solved by toning down the language and not implying that the analysis demonstrates such large claims. Some additional discussion of other factors not considered in the analysis would also be beneficial. Ideally however, a more thorough analysis would be done that takes into account a wider range of factors, including the preceding time period, and compares responses between similarly impacted areas of similar vegetation controlling for other explanations of the dynamics.

Response:
We see the point of the reviewer that we cannot draw a conclusion regarding the legacy effect. This point was also raised by the handling editor. We have therefore removed any statement regarding a potential legacy effect and removed such conclusions throughout the text.

Statistical language and methodology used

There is sometimes inaccurate, confusing or misleading use of statistical language. For example:
L192 'Z-scores less than -1 and more than 1 indicate significant negative and significant positive anomalies beyond normal variability.'
L238 & elsewhere: 'Z-score, values from -1 and 1 are considered normal (within 1 standard deviation of the mean)'
- actually, in a normal distribution (which I think is what we assume here) z-scores of any value are all within the range of normal variability and are by definition 'normal' (this is why it is called the normal distribution. We might sometimes observe an excess of events compared to normal variability, or we might chose to define 'discovery' or a particular feature (e.g. a drought) at a particular statistical level, but these might still be normally distributed. The word 'significant' also has a particular statistical meaning and is normally considered p=5% or so, i.e. 2.5 sigma for a normal distribution, but is dependent on the level of precision required for an experiment. In particular, variations beyond a single standard deviation are not particularly significant, they should occur 32% of the time in a normal distribution of results. As the section is 'statistical data analysis' it is important using precise statistical language for the work. Similarly,
L230 'extremely low soil moisture content (z-score < -1)'
L233 'was affected by extremely dry air and a similar area was affected by extremely dry soil'
again to me extreme implies something a lot less than 1s.d. away (15% of all events if looking at just the lower shoulder). I would call around 1s.d. a fluctuation. I think this also relates a lot to the other specific comment on how we are defining a drought within the paper. There are several uses throughout the paper of wording that is statistically meaningful, but used in an incorrect way (see minor comments). This results in some confusion and doubt on the strength of the

conclusions and results. Overall, greater care and understanding should to be taken in dealing with statistical concepts.

Response:
Please see our detailed response to your previous comment. We define non-drought conditions as normal. When we mention normal throughout the paper, we are not talking about distribution types. We cannot use the term non-drought because we are using a relative matrix that is not only about normal/non-normal moisture conditions, but also determines normal/non-normal vegetation properties (based on GOSIF or NIRv). Thus, we decided to remain with the definition of "normal" condition as we have specified in the paper.
We will correct all other cases where statistical terms are confusing. See our response to the specific comments please.

Additionally some clarity of the statistical treatment of results would be helpful. My understanding of figures 7/8 is that we see a regression of all spatial pixels from 3 different drought periods (2003, 2018, 2022). I think, but it is not clear, that each point is a pixel as well as potentially an 8-day period (or a single JJA mean anomaly or mean JJA value?). If it is mixing both space and time, then I think we get into questions of spatial autocorrelation, and whether the stregth of the relationship is being artificially boosted by this mixing. To understand this, the figure would benefit from an explicit and detailed explanation of what it is showing. If it is all pixels but showing the JJA average (anomaly or real value, also unclear, see specific figures comment) then perhaps it is ok as a spatial regression only.

Response:
Thank you for the query. For Figure 7, each point represents a single JJA anomaly of GOSIF against VPD or SM for each pixel affected by drought, thus it gives an idea of how the spatial differences in GOSIF anomalies are related to VPD or SM. For Figure 8, the boxplot is also derived for each pixel's partial correlation of SM, VPD and GOSIF for JJA period (so based on 8-day data) of each 2003, 2018, and 2022, i.e., it gives an idea about how the temporal (within each summer year) variation of GOSIF relates to VPD and SM. We have modified our statements and clarified our results in the revised version of the manuscript.

Additionally see the minor comments for some examples of dubious statistical practices. Technical comments: L23 we used the ERA5-Land spatial meteorological dataset between 1970 to 2022 to identify conditions with extreme soil and atmospheric dryness. In the methods it says:
L114 'We used Europe-wide (Longitude: 11°W - 32°E; Latitude: 35.8°N -72°N, approximate area of 4.45 million km2) gridded datasets of daily total precipitation (Precip; mm), daily mean air temperature (Tair; °C), daily mean relative humidity (RH; %) and daily mean soil moisture (SM; m3m-3) of topsoil layer (0-7 cm depth), spanning from 2000-2022.'
L118: 'Precip, Tair and RH datasets from the E-OBS v27.0e dataset'
Which is it out of 1970-2022 and 2000-2022? And why do you say ERA5-Land only when elsewhere it says E-OBS?

Response:
For Precip, Tair, and RH (subsequently used to calculate VPD) we used the EOBS v27.0e dataset, and for soil moisture we used the ERA5-Land dataset spanning from 2000-2022. We corrected the statement in the abstract and clarified this in the Methods section of the revised manuscript.

L25: and the OCO-2 solar induced fluorescence (SIF) as an observational proxy
See specific comment on SIF

Response:
We have clarified this statement in the revised version of the manuscript.

L35: Across different forest types, the deciduous broad-leaved forests were most negatively affected by the extreme conditions in 2022, but Evergreen Needle-Leaf Forests (ENF) distributed in northern regions of Europe showed enhanced canopy greening and SIF signals as a benefit of warming.

Perhaps it is not just a benefit of the warming – from figure 2 it appears that they weren't part of the drought area at all. They exhibit a positive SM anomaly and minimal negative VPD anomaly. This begs the question – are they actually areas of drought? And should they be included as data points in the discussion of the drought impacts?

Response:
We revised this statement in the Abstract and wrote: "Across different forest types, the deciduous broad-leaved forests were most negatively affected by the extreme conditions in 2022 given the extent and severity of drought within the distribution range of these forests. The areas dominated by Evergreen Needle-Leaf Forests (ENF) distributed in northern regions of Europe however experienced a positive SM anomaly and minimal negative VPD in 2022 and the forests showed enhanced canopy greening and SIF signals as a benefit of warming."

L38: 'Higher degree of canopy damage in 2022 in spite of less extreme conditions compared to the previous extreme year points to a legacy effect on forest canopies, and a declined forest resilience in response to more frequent drought events.'

Whilst the result is interesting to point out and relevant, I don't think the level of analysis is able to say that it 'points to a legacy effect' or 'a declined forest resilience'. I think it can be mentioned as needing further study in the discussion section, with hypothetical wording such as 'could be due to legacy effects' but, such a strong wording would require further investigation (for example comparing the difference in responses between comparable forests affected by the 2022 drought but not by the 2018 drought, and testing for other factors such as lower forest health prior to the summer). See specific comment.

Response:
We have addressed this point in response to a previous query.

L60 threshold → thresholds

Response:
Corrected.

L71 The hardest-hit areas were Iberian Peninsula
'the Iberian Peninsula'

Response:
Corrected.

L115 'gridded datasets of daily total precipitation (Precip; mm)
Is precipitation data used in the analysis? As far as I can tell only SM and VPD (calculated via RH and Tair) are used.

Response:
Corrected (we did not use the precipitation data).

L117 and daily mean soil moisture (SM; m3m-3) of topsoil layer (0-7 cm depth),
When we are talking specifically about the impact of drought on forests (rather than say e.g. water stress/perturbation), I'm not sure the topsoil alone is sufficient to quantify the magnitude and the response.

Response:
This point is addressed in response to a previous query.

L121: ECMWF… new land component… ERA5 dataset.
As the paper is depends on heatwaves and weather extremes as measured by ERA5, it is worth mentioning somewhere (here or in the discussion) that ERA5 has some limitations specifically when it comes to modelling extreme events such as heatwaves – it tends to underestimate the extent, due to non-dynamic LAI (https://gmd.copernicus.org/articles/16/7357/2023/gmd-16-7357-2023.html ). I don't think there is anything that can be done to mitigate this, but it is worth pointing out that the meteorological extremes are likely larger than shown (which shouldn't affect the main conclusions, but could potentially back up the results further).

Response:
Thank you for the insight and suggesting the paper. We have added the following in the discussion section:

*"It is important to mention that ERA-5 Land datasets have shown to underestimate the extent of 2003, 2010, and 2018 European heatwaves (Duveiller et al., 2023), due to incorporation of non-dynamic leaf area index in their modeling framework. Thus, the SM drought in the drought years of 2003, 2018, and 2022 might be larger than shown in this study, thus making our results rather conservative."*

L125: I think it is worth adding a citation for the VPD, as there are a number of ways to calculate (and is consistent with citing NIRv formula)

Response:
We have added the citation for calculating VPD in the Methods section.

L127: should really say what resampling methodology. For example it is potentially OK to interpolate Tair, but more care should be made with precipitation and soil moisture (better just to take the conservative value). Probably the results would be more robust if sampling of SIF/NIRv were done to 0.1 instead, but I can understand preferring 0.05 for the forest type data.

Response:
Yes exactly, we preferred 0.05° for the forest type data and resampled the Tair, RH, and SM data by interpolation. We removed the mention of the 'Precipitation' data from the manuscript as we did not use the dataset in the end.

L132: some confusion on brackets open and closing. It would be better to simplify use of brackets.

Response:
We simplified the use of brackets in the revised manuscript.

L145: Given the GOSIF dataset is being used, I think the citation should be done in the same way as the NIRv citation (i.e. using the dataset of Li et al. (2019) ). And we explain more about what the product is. See the rest of the specific comment about SIF

Response:
As mentioned in our previous responses, we provided detailed information about the SIF dataset used in the study and included an appropriate citation in the revised version.

L146: SIF signals provide information about physiological response… whilst NIRv signals provide information about the health status of the canopy
Addition of a citation supporting this would be beneficial. And perhaps a line or two about why SIF gives physiological response (i.e. what is SIF?)

Response:

This point is addressed in response to previous comments.

L154: I'm a bit surprised about the inclusion of EBF. There is not much of this in Europe – and you state it is <1%. As it is 7000km2 and you have approximately 5x5km pixels I guess there are around 280 or so pixels in the dataset? Compared to the next smallest, ~3700? An order of magnitude bigger? It cannot really be distinguished on figure 1. The results for EBF in figures 7&8 are not significant… so why did you include it? The figure space used for EBF might be more interesting if it would be used for an 'all' forest-type category instead.

Response:
Although EBF consists of a low number of pixels, this number was not low enough to entirely exclude this forest type from the analysis.

L161 'the selected five years were selected' → 'these five years were selected'

Response:
Corrected.

L161 'only pixels common across the selected five years were selected (Supplementary Fig. 1), and with more than 50% of the 0.05x0.05 pixel area identified as forests (Supplementary Fig. 1)'
L162 The selected forested area in this study covered an area of 907´875 km2 (about 24% of total land area of Europe) (Figure S1).'
'Supplementary Figure 1 Relationship between SIF and NIRv anomalies during 2003, 2018, and 2022 for different forest types.'

I'm slightly confused – Supplementary Figure 1 is SIF vs NIRv by forest type, no? Why is it being referenced here, and what is the relevance? Also there should be consistency in the referencing label (and it probably doesn't need referencing 3 times in two sentences).

Response:
Here we were referring to Figure 1. We have now corrected this error.

Figure 1: I would suggest changing one of the colors of DBF and WSA as it is not so clear which is which.

Response.
We changed the colors in the revised version of the manuscript.

Figure 1: Is there a reasoning for dividing the forest by NEU, CEU, MED? As far as I can tell this division is not referenced anywhere else in the paper, and all of the analysis is done Europe-wide (although a further breakdown of results by location could be interesting)

Response:
Studies that focus on the forests across Europe often distinguish between different regions due to the distinct climate variability and species distributions. In our study we systematically divide the continent following Markonis et al. (2021) (see caption of Figure 1) and we break down our results by location for example in Lines 36-37, 165, 220, 313, 356.

L182: 'we subset our meteorological (Precip, Tair, and VPD), soil moisture (SM)'
Is the Precipitation and Tair data used? As far as I can tell I only see VPD and SM meteo data being used. I understand the Tair is used in the calculation of VPD, along with the RH but then why include Tair here (and not the RH)? And is the Precipitation data used at all in the analysis? If not why is it mentioned?

Response:
Precipitation was not used and Tair was used to estimate VPD. We have corrected this statement and wrote: "*For this purpose, we subset our meteorological (VPD), soil moisture (SM), and vegetation proxy (NIRv and GOSIF) datasets for the months of June, July, August (JJA) which comprised of fourteen 8-day periods, for each forested pixel between 2000 and 2022.*"

190 'We calculated pixel-wise standardized summer anomalies'

Just to be clear for my own understanding – all anomalies shown in the paper are for the full JJA summer? So you take the overall JJA mean SIF, NIRv, VPD and SM and calculate the anomaly relative to the 22 year JJA mean? (as opposed to anomalies of the 14 8-day periods and then averaging the anomaly). For me, the division into 14 8-day periods for the climate data is confusing if we are only looking at JJA results – why is the 8-day division made (I understand that the SIF/NIRv is 8-day, so is it for consistency? In which case say so)? Also is the JJA mean the average of the 8-day periods, or is it done from the daily data?

Response:
Yes, all anomalies of SM, VPD, NIRv, and GOSIF are a JJA summer mean. The hydrometeorological data of VPD and SM were of daily resolution, which was first resampled to 8-day means to match the 8-day of NIRv/SIF, and then all variables were ultimately converted to the summer JJA mean. We have clarified this in the revised version of the manuscript (see Methods section "Meteorological Dataset").

L192: Z-scores less than -1 and more than 1 indicate significant negative and significant positive anomalies beyond normal variability.
See comment on using accurate statistical language

Response:
We addressed this point in response to previous comments.

L200 'and as normal areas if -1 < VPDz < 1 & -1 < SMz < 1.'

see comment on stats. Say 'non-drought areas' or something else

Response:
We decided not to use non-drought because in order to be consistent, we are using the same relative proxy for detecting extremes in the response variables too (in NIRv and GOSIF). Thus, we decided to use the terms "normal" and "extreme".

L220 'while in 2018 we observed the most widespread drought in northern Europe'

See specific comment about drought definitions. Here we see the largest and most widespread VPD and SM anomalies but it is tough to call it drought. Indeed it differs from research (https://nhess.copernicus.org/articles/23/1921/2023/) which suggests it was Central Europe that experienced the largest drought in 2018 (in terms of multiple metrics: soil wetness, precipitation, evaporation etc). Perhaps there should be some discussion on this and methodological differences.

Response:
We revised this statement and wrote: "*In summer 2022, particularly southern regions of Europe experienced the most pronounced increase in atmospheric (z-score > 1) and soil dryness (z-score < -1) (Figure 2) while in 2018 we observed the most widespread VPD and SM anomalies in northern Europe.*"
We also added a brief discussion on the different observed distributions of drought conditions and explained this with the methodological differences.

Figure 3&4: Z-score, values from -1 and 1 are considered normal (within 1 standard deviation of the mean).
See statistical discussion
Shouldn't use the word 'normal' here as it has a specific statistical meaning. You could always say non-drought (assuming we are able to consider this drought – see specific comment), or 'within a single standard deviation fluctuation' or something

Response:
We addressed this point in response to previous comments.

Figure 4: We have 'GOSIF anomaly' but have not introduced what GOSIF is.

Response:
We addressed this point in response to previous comments.

L262 Figures 5&6
In the maps we see a lot of white color on the map, which doesn't seem to be in the scale (which has shades of blue and red). Is the white area not considered – and if so, can we have

an explanation in the caption why? I guess it is because it is not a part of forest areas at 50%, in which case we should include this important information in the figure caption.
Separately, I assume these are JJA SIF/NIRv anomalies – in which case this information should also be added. Indeed, please add all information relevant to understanding the figure.

Response:
The white areas represent non-forested regions and are therefore not considered in the analysis. We clarified this in the figure caption and added all relevant information in the figure caption.

L260: The ENFs however showed positive NIRv anomalies in 2022, also both in terms of magnitude and spatial coverage and % of total area affected (Figure 6).
If I've understood correctly, this should be expected I think. ENF is distributed in an area (Scandinavia) that has higher SM than the long-term mean (i.e. it is not really in a drought).

Response:
We revised this statement to avoid any confusion in interpreting the positive anomalies occurring despite drought. The new sentence reads: "The ENFs showed positive NIRv anomalies in 2022, in terms of magnitude, spatial coverage, and % of total area affected (Figure 6)."
We also mentioned in the Discussion that "Evergreen Needle-Leaf Forests (ENF) distributed in northern regions of Europe were not exposed to extremely dry conditions in 2022".

L271 The anomalies in NIRv and SIF were most correlated across WSAs (mean r2 = 0.62)
I don't think it is statistically sound to take the mean value of R2 between different datasets. The R^2 should be calculated by putting all the data together and calculated together. It's not something that can just be averaged between different groupings.

Response:
We corrected this statement and reported the maximum $r$ value observed in 2018 ($r$= 0.73).
Here $r$ must have been reported and not $r^2$ (which we corrected as well).

L271 The anomalies in NIRv and SIF were most correlated across WSAs (mean r2 = 0.62)
Also, previously we have been using the correlation coefficient (R) and SF1 shows R, whereas now we are talking about R2? Is this intentional, because obviously these are quite different concepts.

Response
We corrected this typo and reported the $r$ value throughout.

Figure 7. See notes on figures
L272: 'With the increase in VPD positive anomalies (i.e., increased atmospheric dryness),…
(Figure 7)' L275: With decrease in soil moisture (i.e., increased soil dryness),...

L281: 'Spatial regression between standardized SIF anomalies with (a) VPD and (b) SM over the drought areas'
So does the figure show SM & VPD anomalies or not? It's not clear, the figure needs introduction, explanation and a good caption. Then explain what results it shows.

Response:
This query was addressed in response to previous comments. We corrected the mismatch between the text and the figure to avoid confusion.

L276 : 'r2 = 34' & 'r2 = 39':
I assume this is meant to be 0.34 and 0.39 , right?

Response:
Yes. Corrected.

But where do the values come from? I don't see them in the figures and they seem to be a referencing the combined value for all the data? In case it is the latter, then I think it would be better to clearly state in a separate sentence where these values come from (i.e. For the full dataset combining all forest types R2 = 0.34.). Given that it is referenced and compared multiple times (indeed more than the separated forest type figure), it might be useful to include a 'all forest types' plot in the figures.

Response:
Yes the values refer to the combined values for all the data. We clarified this in the text.

L278 (mean r2 = 0.48),
See previous comment – is just the mean value of the R2 of the different lines? If so then it's probably incorrect. You can't just average between different models to get a summary statistic of all the models – instead put all the data together and recalculate the R2. I don't think the mean r2 between different fits really means anything as a statistical concept. Feel free to provide a mathematical explanation or literature citation to correct this however.

Response.
We corrected this in response to a previous query.

L279 'and responded most directly to changes in the soil moisture in the WSA'
I guess this backs up the idea that trees have deep roots and the first 7cm of soil is not so important for them. As the woody savanna contains sparse and lower level vegetation and fewer trees, the topsoil is going to have more of an impact (i.e. more grass & short rooted vegetation is going to contribute to the SIF/NIRv of the pixel). As the study is about drought in forests, it makes sense to use more than just the top layer of soil.

Response:
We addressed this point in response to a previous query.

L302 Figure 9. Temporal partial correlation coefficient of SIF with SM and VPD during summer for detected (a) drought areas and (b) normal areas.
Is it SIF and SM or is it SIF anomalies and SM anomalies. Over what time period are the anomalies calculated? What does the word temporal mean here, as previously we were looking at spatial correlations and data. What is the spatial range that this figure covers? See specific figure comments.

Response:
The partial correlation coefficient is estimated using the 8-day GOSIF and SM values and not their anomalies. The word 'temporal' indicates that the correlation was across time (during the summer season of each 2003, 2018, and 2022 year) for each pixel, as correlations can also be across space (which is what is shown in Figure 8). Temporal correlation was calculated using the 14 8-day period GOSIF and SM/VPD data. We clarified this in the revised version of the manuscript.

L320 Although the atmospheric and soil drought were more extensive and severe (indicated by max observed z-score) in 2018 compared to 2022, the negative impact on forests, indicated by declined SIF, was larger in 2022 pointing to a decreased resilience of forests to drought since previous conditions in 2018.

I think that saying that the decline points to decreased resilience is a big stretch. We are comparing two very broad spatio-temporal averages of SIF change and two very broad spatio-temporal averages of air and soil dryness. There is a lot of complexity and patterns within this that the comparison of Europe-averaged SIF change is not able to capture. Sure, it might be because of a decline in resilience, but it also could be because of different areas covered by the drought, or different timespans of the drought, or the conditions in the year preceding the drought, or any number of aspects not considered.

Response:
We removed the statements regarding decreased resilience throughout the text (see our response to the previous query please).

See specific comments.

L353 Our analysis showed that conditions in summer 2022 reduced vegetation functioning across DBFs the most, as it was indicated by declined SIF signals (Figure 5). While deciduous broad-leaved forests were most negatively affected by the extreme conditions in 2022, Evergreen Needle-Leaf Forests (ENF) distributed in northern regions of Europe showed enhanced canopy greening and SIF signals, through benefiting from the episodic warming.

See specific comment on the area of drought considered.

What is said in this section about differences in species adaptation to drought isn't wrong, it's just that the methodology and analysis of the paper is not necessarily demonstrating this. ENF also experienced a lower drought impact (indeed positive SM anomaly) – so why do we think the reason for the difference in the response is physiological?

Probably there needs to be some rewording to remove the implication that this analysis is demonstrating that the forests are responding differently due to their physiology (even if they do have a different physiology). Also there should be a discussion on the fact that the averaged forest response is different due to differences in the drought events happening at a given location (i.e. in some areas the drought is more or less extreme/persistent).

Response:
We reworded this statement following your suggestion (see our response to a similar query before please).

L375 Higher degree of canopy damage that we observed in 2022, despite less severe conditions compared to the previous extreme year, points towards a lasting impact on forest canopies—a sign of decreased forest resilience in the face of more frequent drought events.

This is a very strong conclusion to make from the analysis presented. There are a lot of complicated factors that are not being considered in the discussion.
See specific comment on legacy/resilience.

Response:
As we responded to previous comments, we removed the statements about resilience from the manuscript.

L395 'Despite less severe overall conditions compared to previous extreme years, the observed higher degree of canopy damage in 2022 suggests a declining resilience of forests to drought,'

As above.

Response:
The statements about resilience are removed from the revised version of the manuscript.

**References**

Dogan, S., Berktay, A., & Singh, V. P. (2012). Comparison of multi-monthly rainfall-based drought severity indices, with application to semi-arid Konya closed basin, Turkey. Journal of Hydrology, 470, 255-268.

Jain, V. K., Pandey, R. P., Jain, M. K., & Byun, H. R. (2015). Comparison of drought indices for appraisal of drought characteristics in the Ken River Basin. Weather and climate Extremes, 8, 1-11.

Lal P, Shekhar A, Gharun M, Das NN (2023) Spatiotemporal evolution of global long-term patterns of soil moisture. Science of the Total Environment 867, 161470

Li, X., Xiao, J., Kimball, J.S., Reichle, R.H., Scott, R.L., Litvak, M.E., Bohrer, G., Frankenberg, C. (2020) Synergistic use of SMAP and OCO-2 data in assessing the responses of ecosystem productivity to the 2018 U.S. drought. Remote Sensing of Environment, 251, 112062. https://doi.org/10.1016/j.rse.2020.112062.

Markonis, Y., Kumar, R., Hanel, M., Rakovec, O., Máca, P., AghaKouchak, A., (2021). The rise of compound warm-season droughts in Europe. Science Advances 7. https://doi.org/10.1126/sciadv.abb9668

Mishra, A. K., & Singh, V. P. (2010). A review of drought concepts. Journal of hydrology, 391(1-2), 202-216.

Tsakiris, G., & Vangelis, H. (2004). Towards a drought watch system based on spatial SPI. Water resources management, 18, 1-12.

Wable, P. S., Jha, M. K., & Shekhar, A. (2019). Comparison of drought indices in a semi-arid river basin of India. Water resources management, 33, 75-102.

Zhang J, Xiao J, Tong X, Zhang J, Meng P, Li J, Liu P, Yu P (2022) NIRv and SIF better estimate phenology than NDVI and EVI: Effects of spring and autumn phenology on ecosystem production of planted forests. Agricultural and Forest Meteorology 315, 108819